

# Radiocarbon Measurements of Ecosystem Respiration and Soil Pore-Space CO$_2$ in Utqiaġvik (Barrow), Alaska

Lydia J. S. Vaughn[1,2], Margaret S. Torn[2,3]

[1]Integrative Biology, University of California, Berkeley, Berkeley, CA, 94720, USA
[2]Lawrence Berkeley National Laboratory, Berkeley, CA, 94720, USA
[3]Energy and Resources Group, University of California, Berkeley, Berkeley, CA, 94720, USA

*Correspondence to*: Lydia J. S. Vaughn (lydiajsvaughn@gmail.com)

**Abstract.** Radiocarbon measurements of ecosystem respiration and soil pore space CO$_2$ are useful for determining the sources

of ecosystem respiration, identifying environmental controls on soil carbon cycling rates, and parameterizing and evaluating models. We measured flux rates and radiocarbon contents of ecosystem respiration, as well as radiocarbon in soil profile CO$_2$ in Utqiaġvik (Barrow), Alaska, during the summers of 2012, 2013, and 2014. We found that radiocarbon in ecosystem respiration ranged from +60.5 to -160 ‰ with a median value of +23.3 ‰. Ecosystem respiration became more depleted in radiocarbon from summer to autumn, indicating increased decomposition of old soil organic carbon and/or decreased CO$_2$ production from

fast-cycling carbon pools. Across permafrost features, ecosystem respiration from high-centered polygons was depleted in radiocarbon relative to other polygon types. Radiocarbon content in soil pore-space CO$_2$ varied between -7.1 and -280 ‰, becoming more negative with depth in individual soil profiles. These pore-space radiocarbon values correspond to CO$_2$ mean ages of 410 to 3350 years, based on a steady-state, one-pool model. Together, these data indicate that soil respiration is derived primarily from old, slow-cycling carbon pools, but that total CO$_2$ fluxes depend largely on autotrophic respiration and

heterotrophic decomposition of fast-cycling carbon within the shallowest soil layers. The relative contributions of these different CO$_2$ sources are highly variable across microtopographic features and time in the sampling season. The highly negative $\Delta^{14}$C values in soil pore-space CO$_2$ and autumn ecosystem respiration indicate that when it is not frozen, very old soil carbon is vulnerable to decomposition. Radiocarbon data and associated CO$_2$ flux and temperature data are stored in the data repository of the Next Generation Ecosystem Experiments (NGEE-Arctic) at http://dx.doi.org/10.5440/1364062.


## 1. Introduction

The flux of CO$_2$ from ecosystems to the atmosphere is a critical component of the global carbon budget. This flux is highly heterogeneous in space and time, so extensive datasets are needed to evaluate the carbon balance within ecosystems. Many measurements have been made of soil surface CO$_2$ emissions using soil chambers and eddy covariance towers (e.g., Baldocchi,

2008; Davidson et al., 2002; Norman et al., 1997; Xu and Baldocchi, 2004). Such measurements reveal spatial and temporal patterns in soil and ecosystem respiration that are important for scaling soil carbon emissions across the landscape or identifying drivers of respiration rates (Wainwright et al., 2015). Bulk CO$_2$ fluxes, however, do not provide information on the cycling rates or relative contributions of different soil carbon pools to total CO$_2$ emissions. CO$_2$ emitted from the soil surface includes rapidly cycling carbon in autotrophic respiration as well as heterotrophic decomposition of soil carbon that cycles on broad range of

timescales. Shifts in these carbon pool distributions can have large long-term consequences for soil carbon stocks, but may be impossible to detect in bulk CO$_2$ flux rates (Hopkins et al., 2012; Schuur et al., 2009; Torn et al., 2009; Trumbore, 2000, 2009).



Linking $CO_2$ emissions with carbon cycling rates requires a tracer of carbon dynamics that can differentiate between source carbon pools. Natural abundance radiocarbon provides such a tracer, as the radiocarbon content of $CO_2$ reflects the age (since photosynthesis) and decomposition rates of its component sources (Trumbore, 2000). Accordingly, radiocarbon in surface $CO_2$ emissions and soil pore gas can be used to quantify carbon cycling rates and assess their variability across space, time, and

environmental factors such as thaw depth, soil moisture, temperature, and vegetation (Gaudinski et al., 2000; Trumbore, 2000). Within and across sites, such variations can indicate differences in substrate utilization by microbial decomposers (Borken et al., 2006; Chasar et al., 2000) and shifts between more fast-cycling and slow-cycling substrate pools (Hicks Pries et al., 2013; Hopkins et al., 2012; Schuur et al., 2009). Radiocarbon measurements of soil respiration may be particularly useful for models of the carbon cycle, as a means to determine parameters and evaluate model performance. Similarly, the radiocarbon signature

of respiration can be used to constrain the terrestrial signal in top-down carbon cycle analyses (He et al., 2016; Randerson et al., 2002).

In the high latitudes in particular, increased decomposition rates from large, slow-cycling soil carbon pools have the potential to generate an important long-term climate change feedback (Schuur et al., 2015). There, an estimated 1300 Pg of soil carbon has been protected from decomposition by cold temperatures and often frozen or anoxic conditions (Hugelius et al., 2014), factors

that are expected to change as climate change warms soils, alters hydrology, and degrades permafrost (ACIA, 2004). In the lower depths of the seasonally thawed active layer, carbon that has historically cycled on millennial timescales may be highly decomposable under aerobic and thawed conditions (Mueller et al., 2015; Strauss et al., 2014; Waldrop et al., 2010). Our current understanding of this old carbon's decomposability, however, is based primarily on $CO_2$ production rates from laboratory incubations, which may not accurately reflect in situ decomposition dynamics. To quantify in situ decomposition rates, field

radiocarbon measurements can be used to differentiate between slow-cycling and fast-cycling carbon. When measured across seasonal and environmental gradients, radiocarbon abundances in soil respiration link these decomposition dynamics to their environmental controls.

In spite of their use in tracing carbon cycle dynamics, only a limited set of radiocarbon data have been published for Arctic ecosystem respiration (e.g., Hardie et al., 2009; Hicks Pries et al., 2013; Lupascu et al., 2014; Phillips et al., 2015) or soil pore-

space $CO_2$ (Czimczik and Welker, 2010; Lupascu et al., 2014a, 2014b), from only a few sites and tundra types. To our knowledge, no data are currently available on radiocarbon in $CO_2$ from Arctic polygon tundra. Here, we present measurements of radiocarbon in ecosystem respiration and soil pore-space $CO_2$ from a polygon tundra ecosystem in Utqiaġvik, Alaska. We measured rates and radiocarbon contents of soil surface $CO_2$ emissions across three summer seasons, from a range of ice wedge polygon features. Using soil pore-space $^{14}CO_2$ profiles, we quantified vertical distributions of carbon cycling rates within the soil

profile. Together, these measurements reveal temporal patterns and spatial relationships in ecosystem respiration and its source carbon pools.

## 2. Study Site

Field sampling was conducted at the Barrow Environmental Observatory (BEO), ~6 km east of Utqiaġvik (formerly Barrow), Alaska (71.3 °N, 156.5 °W), at the northern end of the Alaskan Arctic coastal plain. Utqiaġvik has a mean annual temperature of

-12 °C and mean annual precipitation of 106 mm, with long, dry winters and short, moist, cool summers. The land surface has low topographic relief reaching a maximum elevation of 5 m (Brown et al., 1980; Hubbard et al., 2013). The seasonally-thawed active layer ranges from 20 to 60 cm, underlain by continuous ice-rich permafrost to depths greater than 400 m (Hinkel and



Nelson, 2003). Formed from the late Pleistocene Gubic formation (Black, 1964), soils in the region are dominated by Typic Aquiturbels (53 %), Typic Histoturbels (22 %), and Typic Aquorthels (8.6 %) (Bockheim et al., 1999).

Within the BEO, ~65 % of the ground surface is covered by ice wedge polygons (Lara et al., 2014), discrete landscape units formed by the growth and degradation of subsurface ice wedges (Billings and Peterson, 1980; Brown et al., 1980). Individual

polygons are 10-20 m in diameter with raised, relatively dry rims at their perimeter and are separated by low-lying, saturated troughs. Polygons can be classified according to the elevation of their centers; low-centered polygons (LC) have standing water and primarily graminoid vegetation while high-centered polygons (HC) have dry surface soils and a greater abundance of mosses and lichens. Between HC and LC polygons, flat-centered polygons (FC) have intermediate morphology and subsurface properties. Among LC, FC, and HC polygons, differences in active layer depth, oxygen availability, soil carbon distribution, and

thermal conductivity (Liljedahl et al., 2016; Lipson et al., 2012; Ping et al., 1998) create strong differences in surface carbon fluxes (Vaughn et al., 2016; Wainwright et al., 2015). To capture this range of microtopography and associated carbon cycle controls, we collected $CO_2$ samples from centers, rims, and troughs of the three polygon types.

### 3. Field Measurements and Sample Collection

### 3.1 Surface $CO_2$ Emissions

In August and October 2012, July and September 2013, and September 2014, we collected soil surface $CO_2$ emissions for radiocarbon analysis using a static chamber method modified from Hahn et al. (2006). Samples were collected from a total of 19 locations within 11 polygons (4 LC, 4 FC, 3 HC). Opaque chambers (25 cm diameter) were seated on circular PVC chamber bases extending to depths of ~10 cm below the soil surface. We installed all bases at least two days prior to sampling to limit the influence of disturbance on gas flux rates and radiocarbon values. Based on variations in the surface and depths of bases,

aboveground chamber height varied between ~15 and 20 cm. Chambers blocked light transmission and were tall enough to enclose surface vegetation, so $CO_2$ emissions were equivalent to ecosystem respiration. For sample collection, the chamber was placed in a 3 cm-deep channel on the top rim of each base, which was filled with water to create an airtight seal. We then circulated chamber gas through soda lime for 20 minutes at a flow rate of 1 L min$^{-1}$ to remove ambient $CO_2$. $CO_2$ was allowed to accumulate in the chamber over 2 to 48 hours, depending on the rate of $CO_2$ accumulation, which we monitored periodically by

passing a 30 mL sample of chamber gas through a LI-820 $CO_2$ gas analyzer (LI-COR) at a flow rate of ~1 L min$^{-1}$. For all samples, the final chamber $CO_2$ concentration was more than twice its initial concentration, important for accuracy in chamber radiocarbon measurements (Egan et al., 2014). Based on this concentration measurement, a volume of chamber gas sufficient for radiocarbon analysis was collected in one or more 500-1000 mL evacuated stainless steel canisters connected by capillary tubing to a chamber sampling port. High-concentration samples were collected with a syringe and needle through a septum in the

sampling port and immediately injected into evacuated glass vials sealed with 14 mm-thick chlorobutyl septa (Bellco Glass, Inc.). To correct chamber gas samples for atmospheric contamination, we collected local air samples in 3000 mL stainless steel canisters on August 12, 2012, July 13, 2013, and September 2 and 7, 2014.

In July and September 2013 and September 2014, we measured rates of ecosystem respiration from radiocarbon sampling locations. $CO_2$ fluxes were measured within 2 days of radiocarbon sample collection, using opaque static chambers seated on

bases described above and vented according to Xu et al. (2006) to minimize pressure changes due to the Venturi effect. For each measurement, we measured $CO_2$ concentrations within the chamber over a period of 4-8 minutes using a Los Gatos Research, Inc. Portable Greenhouse Gas Analyzer. We calculated the $CO_2$ flux rate (equivalent to ecosystem respiration) as the slope of



the linear portion of its concentration vs. time curve, converted to units of µmol m$^{-2}$ s$^{-2}$ according to chamber volume and temperature. Endpoints of this linear region were determined manually for each curve, and measurements lacking a clear linear range were not included in the dataset. Corresponding with each $CO_2$ flux measurement, we measured thaw depth with a tile probe.

**3.2. Soil Pore-Space $CO_2$**

In August 2012 and July 2013, we collected soil pore gas from a total of 6 soil profiles within 5 polygons (1 LC, 2 FC, 2 HC). Samples were collected with a method similar to Czimczik and Welker (2010). Briefly, 1/4" diameter stainless steel probes were inserted into the soil at 45 ° angles to vertical depths of 10, 20, and 30 cm, or to 2 cm above the frost table if thaw depth was less than 30 cm. Probes were capped with gastight septa and allowed to remain in place throughout the sampling season. Before

collecting each sample, we purged 10 mL of gas from the probe and measured the $CO_2$ concentration with a LI-820 $CO_2$ gas analyzer as described above. Radiocarbon samples were collected by connecting evacuated 500-1000 mL stainless steel canisters to probes via flow-restricting tubing (Upchurch scientific, 0.01" ID × 10 cm length), which allowed canisters to fill slowly over 4 hours with minimal disturbance to the soil $CO_2$ concentration gradient (Gaudinski et al., 2000). In-line Drierite water traps were used during sample collection to prevent moisture accumulation in canisters. As with surface respiration

samples, high-concentration samples were collected from probes with a syringe and needle and immediately injected into evacuated glass vials. Due to water-saturated soils or clogged soil probes, we were unable to obtain samples from all profiles and depths. For this reason, the final sample set represents only a subset of depths and sampling locations.

**3.4. Sample Purification and Radiocarbon Analysis**

$CO_2$ from gas samples was cryogenically purified under vacuum, divided for $^{14}C$ and $^{13}C$ analysis, and sealed in 9 mm quartz

tubes. For radiocarbon analysis, we sent samples to Lawrence Livermore National Laboratory's Center for Accelerator Mass Spectrometry (CAMS) or the Carbon, Water, and Soils Research Lab at the USDA-FS Northern Research Station, where $CO_2$ was reduced to graphite on iron powder under $H_2$. $^{14}C$ abundance was then measured at CAMS using an HVEC FN Tandem Van de Graaff accelerator mass spectrometer or at UC Irvine's Keck Carbon Cycle AMS facility. $^{13}C/^{12}C$ in $CO_2$ splits was analyzed on the UC Davis Stable Isotope Laboratory GVI Optima Stable Isotope Ratio Mass Spectrometer.

Following the conventions of Stuiver and Polach (1977), radiocarbon results are presented as fraction modern relative to the NBS Oxalic Acid I (OX1) standard ($F^{14}C$), and deviations in parts per thousand (‰) from the absolute (decay-corrected) OX1 standard ($\Delta^{14}C$). All results have been corrected for mass-dependent isotopic fractionation using $^{13}C$ measurements.

**4. Calculations and Data Quality Control**

We corrected surface-chamber radiocarbon measurements for atmospheric contamination using the method described in Schuur

and Trumbore (2006). Briefly, we determined fractional contributions of background atmosphere and ecosystem respiration to total chamber gas using $^{13}C$ values in a two-pool mixing model:

$$^{13}C_S = f_{Reco} \times ^{13}C_{Reco} + f_{atm} \times ^{13}C_{atm} \, , \qquad (1)$$

$$f_{Reco} + f_{atm} = 1 \, , \qquad (2)$$





where, $f_{\text{Reco}}$ and $f_{\text{atm}}$ are the fractional contributions of ecosystem respiration and background atmosphere, $^{13}\text{C}_\text{S}$ and $^{13}\text{C}_\text{atm}$ are the measured $^{13}\text{C}$ abundances in the sample and background atmosphere in units of atom %, and $^{13}\text{C}_\text{Reco}$ is the $^{13}\text{C}$ abundance in ecosystem respiration, approximated separately for each polygon type as the mean $^{13}\text{C}$ of chamber $\text{CO}_2$ samples with $[\text{CO}_2] >$ 4000 ppm. To minimize error due to large proportions of atmospheric $\text{CO}_2$, we omitted samples with $f_{\text{Reco}} < 0.5$. For each

sample, we calculated $\Delta^{14}\text{C}$ of ecosystem respiration ($\Delta^{14}\text{C}_\text{Reco}$) according to Eq. (3):

$$\Delta^{14}\text{C}_\text{S} = f_{\text{Reco}} \times \Delta^{14}\text{C}_\text{Reco} + f_{\text{atm}} \times \Delta^{14}\text{C}_\text{atm} , \tag{3}$$

where $\Delta^{14}\text{C}_\text{S}$ and $\Delta^{14}\text{C}_\text{atm}$ are the measured $\Delta^{14}\text{C}$ values of the sample and background atmosphere, and $f_{\text{Reco}}$ and $f_{\text{atm}}$ were calculated from Eq. (1) and Eq. (2). Errors due to analytical precision and variations in source isotopic signatures were propagated through this correction according to the formulation in Phillips and Gregg (2001).

Soil surface $\text{CO}_2$ flux measurements were assessed for quality using two criteria, the standard error of the slope (SE) and the per cent relative standard error (PRSE), defined as $100 \times \text{SE}_{\text{slope}} / \text{Estimate}_{\text{slope}}$. Flux measurements with SE > 0.05 and PRSE > 5 were omitted from the dataset. This set of dual criteria avoided biasing the dataset toward low fluxes (if SE alone were used) or high fluxes (if PRSE alone or $R^2$ were used).

With soil pore-space radiocarbon data, we omitted measurements with $\text{CO}_2$ concentrations less than 400 ppm due to possible
leakage and atmospheric contamination during sampling, with the exception of one sample from 31 cm depth with highly negative $\Delta^{14}\text{C}$, indicating a low proportion of atmospheric $\text{CO}_2$. At this field site, $^{13}\text{C}$ abundances in pore-space $\text{CO}_2$ vary greatly due to isotopic fractionation from methane production and consumption (Vaughn et al., 2016). For this reason, we could not correct subsurface samples for atmospheric $\text{CO}_2$. Reported radiocarbon values thus represent the total $\text{CO}_2$ present in the soil pore-space, sourced from heterotrophic respiration, root respiration, and downward atmospheric diffusion.

With soil-profile $\text{CO}_2$ samples, we used radiocarbon measurements to model the mean age of carbon in respired $\text{CO}_2$ using the time-dependent steady state turnover time model described in Torn et al. (2009), modified to account for carbon residence time in plant tissues:

$$F'_{\text{C},t} C_t = I F'_{\text{atm},t-\text{T}_\text{R}} + C_{t-1} F'_{\text{C},t-1}(1 - \tfrac{1}{\tau} - \lambda) , \tag{4}$$

where:

$F' = \frac{\Delta^{14}C}{1000} - 1$

$F'_\text{C} = F'$ of the given carbon pool, equal to $F'$ of the $\text{CO}_2$ sample

$F'_\text{atm} = F'$ of $\text{CO}_2$ in the local atmosphere

$I$ = input rate of carbon from the atmosphere to the given carbon pool (g C y$^{-1}$)

$C$ = stock of carbon in the given carbon pool (g)

$\tau$ = turnover time of the given carbon pool, equivalent to the mean age of carbon in its decomposition flux (y)

$\lambda$ = radioactive decay rate of $^{14}\text{C}$ (1/8267 y)

$\text{T}_\text{R}$ = mean residence time of carbon in plants before entering soil organic matter (y).



At steady state, $C_t = C_{t-1} = I \times \tau$, so Eq. (4) reduces to:

$$F'_{C,t} = \frac{1}{\tau}F'_{\text{atm},t-T_R} + F'_{C,t-1}\left(1 - \frac{1}{\tau} - \lambda\right). \tag{5}$$

Following Eq. (5), the $\Delta^{14}C$ value of $CO_2$ at time $t$ thus depends on the turnover time of carbon in the decomposing carbon pool, the mean residence time of carbon in plant material, and the $\Delta^{14}C$ of atmospheric $CO_2$ in the current and previous year, which has

changed continuously since the release of radiocarbon into the atmosphere from nuclear weapons testing between 1950 and the mid 1960s (Trumbore, 2000). This model assumes that $CO_2$ is derived from a homogeneous pool of decomposing carbon, such that the turnover time of this pool is equal to the mean age of its decomposition flux (Sierra et al., 2017).

The mean residence time of carbon in plants reflects a mixture of materials with varying transfer rates. Some photosynthates enter the soil within 1 day of fixation (Loya et al., 2002), whereas carbon resides in longer-lived plant organs from ~2 to as long

as 15 years before entering the soil organic matter pool (Billings et al., 1978; Dennis, 1977). We assumed that across plant organs, plant species, and seasons, the mean value of $T_R$ lies between 0 and 5 years.

Annual atmospheric $\Delta^{14}C$ values were compiled from our data and three other sources: the IntCal13 dataset (Reimer et al., 2013), measurements from Fruholmen, Norway between 1962-1991 (Nydal and Lövseth, 1996), measurements from Utqiaġvik between 1999-2007 (Graven et al., 2012), and our measurements from the BEO in 2012, 2013, and 2014 (Table S1, Fig. S1). From

roughly the same latitude as Utqiaġvik, Fruholmen $\Delta^{14}C$ measurements provide a close approximation for missing Utqiaġvik data (Meijer et al., 2008). Because ecosystem $CO_2$ uptake occurs primarily during the growing season, we averaged June-August $\Delta^{14}C$ values to produce an annualized summer dataset. Data gaps from 1992-1998 and 2008-2011 were filled using an exponential interpolation constrained by the available data from the 10 years surrounding each gap.

Using our measured $\Delta^{14}C$ values and annually resolved atmospheric $\Delta^{14}C$ data, we iteratively solved for the mean age of each

$CO_2$ sample. We performed this calculation twice, using $T_R$ values of 0 and 5 to bracket the likely $T_R$ range. Samples containing a large percentage of recently fixed carbon yielded two possible solutions for each $T_R$ value (Trumbore, 2000). In such cases, we chose the appropriate solution either by comparing the two values to other (unique) mean age values within the same profile, or by comparing the measured carbon stock with the carbon stocks calculated from the $CO_2$ production rate and the candidate solutions (Torn et al., 2009).

**3. Results and Discussion**

**3.1 Radiocarbon in Ecosystem Respiration**

Radiocarbon contents of soil surface $CO_2$ emissions, equivalent to ecosystem respiration ($\Delta^{14}C_{\text{Reco}}$), ranged from +60.5 to -160 ‰, with a median value of +23.3 ‰ (Table 1). The positive $\Delta^{14}C_{\text{Reco}}$ values measured in 28 of the 37 samples indicate high proportions of carbon fixed since 1950, likely sourced from both autotrophic respiration and decomposition of rapidly cycling

soil carbon in shallow soil layers. In contrast, the negative values measured in the other 9 samples show that the sources of $CO_2$ were dominated by carbon that cycles on centennial to millennial timescales. $\Delta^{14}C_{\text{Reco}}$ followed a left-skewed distribution, with notably negative $\Delta^{14}C_{\text{Reco}}$ values measured from LC1-center in October 2012 (-159.5 ‰) and HC1-center in September 2013 (-115.5 ‰). The variations in these $\Delta^{14}C_{\text{Reco}}$ values reflect differences in the relative $CO_2$ production and transport rates from both autotrophic and heterotrophic source pools, whose decomposition rates and radiocarbon contents vary spatially and temporally





(Nowinski et al., 2010). Together, seasonal variations in soil thaw depths and spatial variations in soil organic carbon $^{14}$C depth profiles affect the radiocarbon signature of surface $CO_2$ emissions.

We observed a general decrease in $\Delta^{14}C_{Reco}$ as the sampling season progressed (Fig. 1). This seasonal decline in $\Delta^{14}C_{Reco}$ is consistent with other data from high latitude sites (Hicks Pries et al., 2013; Trumbore, 2000) and reflects seasonal changes in

thaw depth, soil temperature, and vegetation activity. Surface soils at this site begin to thaw in June, typically reaching their maximum temperatures in July (Hinkel et al., 2001; Torn, 2015). As shallow soils warm and plant activity increases in this early summer period, ecosystem respiration includes high proportions of $^{14}$C-enriched $CO_2$ from autotrophic respiration and heterotrophic decomposition of shallow, rapid-cycling soil carbon. Later in summer and into the autumn, the balance of respiration shifts toward increased importance of deeper soil decomposition. Autotrophic respiration peaks in July or August,

and decreases substantially into the fall after plants senesce (Hicks Pries et al., 2013). During the autumn season, surface soils refreeze while deep soils continue to warm (Table 1) (Zona et al., 2016), limiting heterotrophic respiration from shallow soils while enhancing decomposition from deeper, more $^{14}$C-depleted soil carbon pools. The effect of these changes is a seasonal shift in respiration from primarily shallow, fast-cycling source carbon pools to more deep, $^{14}$C-depleted soil organic matter.

Across time-series measurements from individual profiles or polygon types, $\Delta^{14}C_{Reco}$ varied not only with time in the sampling

season but also with polygon type (Table 1, Fig. 1). HC polygons displayed particularly strong seasonal trends in $\Delta^{14}C_{Reco}$; in each HC polygon from which multiple measurements were made, $\Delta^{14}C_{Reco}$ decreased throughout the summer months to a minimum value in September or October. The magnitude of this seasonal decrease varied greatly among individual HC polygons and microtopographic positions within the polygons, reaching a minimum $\Delta^{14}C_{Reco}$ value of -115 ‰ from the center of polygon HC1 but only +13 ‰ from the trough of HC3 (Table 1). In contrast with HC polygons, $\Delta^{14}C_{Reco}$ from FC and LC

polygons remained closer to atmospheric values throughout the sampling season (Fig. 1). One exception was one highly negative $\Delta^{14}C_{Reco}$ value measured in October from the center of polygon LC1. In this instance, with October thaw depths close to their annual maximum, thaw may have penetrated into the transition layer at the top of the permafrost, exposing old, previously frozen carbon to decomposition. Alternatively, this isolated measurement of $^{14}$C-depleted $CO_2$ may reflect heterogeneity within the active layer due to cryoturbation (Kaiser et al., 2007; Ping et al., 1998).

The influence of microtopography on old carbon emissions was particularly apparent in September 2014 (Fig. 2). At this time, decomposition of old carbon consistently dominated the respiration flux from HC polygon centers ($\Delta^{14}C_{Reco}$ = -51.6 ± 32.4 ‰), whereas fast-cycling pools dominated respiration from FC and LC polygon centers (13.0 ± 10.9 ‰ and 5.9 ± 5.6 ‰ respectively). Unlike $\Delta^{14}C_{Reco}$, concurrent ecosystem respiration rates were comparable across the three polygon types (Fig. 2b), which indicates that absolute rates of old carbon decomposition were greater from HC polygons than from LC or FC polygons.

Interestingly, soil thaw at this time was deepest in FC polygons (Table 1); across profiles at the site scale, we saw no correlation between thaw depth and $\Delta^{14}C_{Reco}$. This finding suggests that the relationship between the depth of thaw and old carbon mineralization depends on the scale of observation. At the scale of an individual profile, seasonal variations in $\Delta^{14}C_{Reco}$ correspond with changes in thaw depth. At the site scale, however, thaw depth may not be a useful predictor of spatial variations in $\Delta^{14}C_{Reco}$.

In July and September 2013 and September 2014, ecosystem respiration ranged between 0.32 and 2.5 µmol $CO_2$ m$^{-2}$ s$^{-1}$ (Table 1). The relationship between the rate and radiocarbon content of ecosystem respiration was highly variable (Fig. 3), but with two notable patterns across all data. As ecosystem respiration increased, the variance of $\Delta^{14}C_{Reco}$ tended to decrease, and there were



no negative $\Delta^{14}C_{Reco}$ measurements associated with fluxes above 1 μmol $CO_2$ m$^{-2}$ d$^{-1}$ (Fig. 3). These patterns reflect the sensitivity of ecosystem respiration to the most dynamic carbon sources, autotrophic respiration and shallow soil carbon mineralization. Autotrophic respiration can contribute as much as 70 % of ecosystem respiration at the peak of the growing season (Hicks Pries et al., 2013), declining markedly as plants senesce. Similarly, heterotrophic decomposition rates are both

highest and most temporally variable in shallow soil layers (Hicks Pries et al., 2013), where soil temperatures exhibit a large seasonal range. As a result, old, slow-cycling carbon from deep soil respiration comprises a large percentage of the total carbon flux only when ecosystem respiration rates are low.

### 3.2 Radiocarbon in Soil Pore-Space $CO_2$

At a subset of locations and sampling dates, we measured the radiocarbon content of $CO_2$ in soil pore gas ($\Delta^{14}C_{CO2p}$). $\Delta^{14}C_{CO2p}$

became increasingly negative with depth in the soil, but varied widely among profiles and sampling dates from -7.1 to -280 ‰ (Table 2, Fig. 4). These negative values indicate that pore-space $CO_2$ was derived primarily from older soil organic matter, with minimal contributions from plant-respired carbon or fast-cycling soil organic carbon. In general, ecosystem respiration was enriched in radiocarbon relative to soil pore-space $CO_2$, even at only 10 cm depth (Table 1, Table 2). This observation suggests that the shallowest $CO_2$ sources—autotrophic respiration and/or heterotrophic decomposition of fast-cycling (annual-decadal)

organic carbon—contribute large proportions of the total respiration flux, even late in the season when plants have largely senesced. Detecting and characterizing the decomposition of older, deeper soil organic carbon requires direct measurements of soil pore-space $CO_2$.

In three soil profiles (HC3-trough, HC3-center, and FC2-center), respired $CO_2$ became enriched in radiocarbon near the permafrost table (Fig. 2), a pattern that has been previously observed in the Arctic (Lupascu et al., 2014a). In these three

profiles, concentrations of $CO_2$ in deep pore-space samples were 6 to 25 times higher than background atmosphere, so we infer that the higher $\Delta^{14}C_{CO2p}$ values at depth were not caused by downward transport of atmospheric $CO_2$. Instead, this pattern indicates significant $CO_2$ production from fast-cycling carbon near the permafrost table. As in other Arctic sites, this depth trend is likely due either to cryoturbation (Bockheim and Tarnocai, 1998) or DOC leaching (Lupascu et al., 2014a), both of which can transport recently fixed, relatively decomposable carbon to the deep section of the active layer.

With $\Delta^{14}C_{CO2p}$ values, we used a single pool turnover time model to calculate the mean age of pore-space $CO_2$ (Table 2). This model assumes that $CO_2$ is respired from a homogeneous pool of decomposing carbon (Trumbore, 2000) such that the turnover time equals the mean age of the respired carbon (Sierra et al., 2017). Across all measurement dates and depths, we found the mean age of respired $CO_2$ ranged from 410 y in one 10 cm sample to 3350 y at only 20 cm depth, indicating decomposition of old and/or slow-cycling carbon, even in relatively shallow soils. HC polygon centers produced particularly old $CO_2$, with ages

greater than 3000 y from HC3-center in July 2013 and HC1-center in August 2012 and September 2013. This observation aligns with the spatial patterns we observed in surface $CO_2$ emissions, which were particularly $^{14}C$-depleted in HC polygons (Table 1, Fig. 1). Together, these findings suggest that low $\Delta^{14}C_{Reco}$ from HC polygon centers was due to mineralization of very old carbon at depth.

Our findings of consistently negative $\Delta^{14}C$ values from soil pore-space $CO_2$ are similar to those from the other Arctic sites where

it has been measured (Czimczik and Welker, 2010; Lupascu et al., 2014b, 2014a). In contrast, studies in temperate or tropical sites find $\Delta^{14}C_{CO2p}$ values that are more similar to atmospheric values (Borken et al., 2006; Gaudinski et al., 2000; Trumbore, 2000), where soil respiration is dominated by root respiration and decomposition of rapidly cycling soil organic matter. This



difference may be due in part to differences among biomes in rooting distributions. Because of strong vertical gradients in soil temperature and nutrient and water availability, rooting distributions in Arctic tundra are extremely shallow (Iversen et al., 2015), limiting most root respiration to the shallow organic layer. Additionally, such negative $\Delta^{14}C_{CO2p}$ values suggest that old, slow-cycling organic matter in Arctic soils is readily decomposable under thawed conditions. Cold soil temperatures and frozen

conditions limit microbial activity throughout much of the year, allowing otherwise-decomposable soil organic matter to accumulate. During the short season in which Arctic soils are thawed, these large pools of $^{14}$C-depleted soil carbon may be largely unprotected against microbial mineralization. Compared with tropical or temperate soils, existing Arctic soil carbon stores may thus be particularly vulnerable to soil thaw and active layer deepening due to climate change.

Due to sampling limitations, soil profile $^{14}$C data are available from only a subset of polygon types and positions. No samples

were collected from polygon rims, and only few samples were obtained where soils were saturated (Table 2). For this reason, our dataset does not capture microtopographic variations in deep soil decomposition rates and controls. Soil temperature profiles, vegetation, and soil pore-space oxygen availability influence microbial activity and vary among profiles and with time in the thawed season (Lipson et al., 2012; Olivas et al., 2011). To characterize the relationships between these variables and soil carbon decomposition rates, further measurements are needed of $^{14}$C in soil pore-space DIC and $CO_2$, across spatial, seasonal,

and hydrological gradients.

**Data Availability**

Data are stored in the Next-Generation Ecosystem Experiments (NGEE-Arctic) data repository (Vaughn et al., 2018) and may be accessed at http://dx.doi.org/10.5440/1364062. The code used to generate plots and correct chamber samples for atmospheric contamination can be found at https://github.com/lydiajsvaughn/Radiocarbon_field_Barrow. The code used for turnover time

modeling can be found at https://github.com/lydiajsvaughn/Radiocarbon_inc_2012.

**Conclusions**

We measured the radiocarbon contents of ecosystem respiration and soil pore-space $CO_2$ between 2012 and 2014 in Utqiaġvik, Alaska. As a naturally occurring tracer of soil carbon dynamics, radiocarbon in $CO_2$ and soil carbon reflects the rates at which carbon cycles through plant and soil pools (Trumbore, 2000). Radiocarbon provides a powerful tool to test and parameterize

models (He et al., 2016) and evaluate how environmental variables, seasonal changes, and disturbance influence decomposition rates from fast- and slow-cycling carbon pools. In Arctic sites, where cycling rates can be very slow, radiocarbon can be used to detect changes in decomposition from millennial-cycling pools (Hicks Pries et al., 2013; Schuur et al., 2009), which strongly influence long-term carbon dynamics but may be impossible to detect with other methods. Only limited data are available of radiocarbon in ecosystem respiration or soil pore-space $CO_2$, particularly from Arctic sites where samples can be difficult to

obtain due to low rates of $CO_2$ production. To characterize seasonal and spatial variations in the age of carbon pools contributing to total $CO_2$ efflux, we measured radiocarbon in ecosystem respiration and soil pore-space $CO_2$ across three summer seasons in Arctic polygon tundra.

$\Delta^{14}$C in ecosystem respiration varied between +60.5 and -160 ‰ across the three years, with a strong seasonal trend (Fig. 1). July and August $\Delta^{14}C_{Reco}$ measurements were generally close to the $\Delta^{14}CO_2$ values of the local atmosphere, which declined from 22.2

‰ in 2012 to 17.7 ‰ in 2014. Later in the season, we measured $\Delta^{14}CO_2$ values that differed greatly from the local atmosphere (Fig. 1), contributing $^{14}$C-depleted $CO_2$ to the atmospheric pool. These seasonal variations in $\Delta^{14}C_{Reco}$ and respiration rates



contribute to the strong seasonal cycles of atmospheric $\Delta^{14}CO_2$ observed in the high latitudes (Graven et al., 2012). By quantifying these variations—respiration flux rates and radiocarbon values over space and time—this dataset offers useful information for atmospheric budgeting and inversions.

Measurements of soil pore-space $CO_2$ and late-season ecosystem respiration indicate that carbon that cycles on millennial timescales contributes substantially to soil respiration. $\Delta^{14}C_{CO2p}$ declined steeply with depth, particularly in high-centered polygons (Fig. 3); at only 20 cm below the soil surface, the mean age of carbon in the decomposition flux was as old as 3000 y (Table 2). When thaw depth approached its maximum in September and October, highly depleted $^{14}C$ in respiration indicated that carbon older than 1000 years was a major source of heterotrophic respiration. Together, these observations suggest that during the short summer thaw season, ancient carbon stores become available to decomposition, stabilized otherwise by cold,
often frozen, and often anaerobic conditions (Mikan et al., 2002; Schmidt et al., 2011; Schuur et al., 2015).

As climate change alters these environmental controls and soils warm and thaw, a key question is how decomposition rates will change. A particularly important—but unknown—factor is the decomposition rate of carbon released from thawing permafrost (Hicks Pries et al., 2013; Koven et al., 2015; Kuhry et al., 2013). Our measurements cannot differentiate between newly thawed soil organic matter and carbon that has historically experienced an annual thaw. Our data do show, however, that when it is
unfrozen, even very old Arctic soil organic carbon will readily decompose. As thaw depth progressed to the permafrost boundary, we consistently documented $CO_2$ production via decomposition of centennial carbon pools. As permafrost thaw progresses and Arctic soils warm, further measurements of soil-respired $^{14}CO_2$ across spatial and temporal gradients will provide critical information on soil carbon vulnerability.

**Author contribution**

LJSV and MST designed the sampling procedure, LJSV collected and analysed data with assistance and guidance from MST, and LJSV prepared the manuscript with contributions from MST.

**Competing interests**

The authors declare that they have no conflict of interest.

**Acknowledgments**

We thank J. Bryan Curtis and Oriana Chafe for field sampling assistance, and the Radiocarbon Collaborative for radiocarbon analyses. This research was conducted through the Next Generation Ecosystem Experiments (NGEE-Arctic) project, which is supported by the Office of Biological and Environmental Research in the US Department of Energy Office of Science.

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



**Table 1. Flux rate and radiocarbon abundance of soil surface CO$_2$ emissions with associated thaw depth and temperature measurements**

| Profile | Date | Thaw depth (cm) | Air temp °C | 5cm soil temp* °C | 10cm soil temp* °C | CO$_2$ flux** (µmol m$^{-2}$ d$^{-1}$) | $^{14}$C Analysis year | $F^{14}$C | $\Delta^{14}C_{CO2}$*** (‰) |
|---|---|---|---|---|---|---|---|---|---|
| HC1-center | 8/9/12 | 31 | -- | 5.9 ± 0.0 | 4.3 ± 0.0 | -- | 2013 | 1.0298 | -58.4 ± 6.1 |
| | 9/2/14 | 32 | 4.6 | 4.3 ± 0.6 | 3.3 ± 0.5 | 0.547 ± 0.0047 | 2016 | 0.8916 | -115.5 ± 10.2 |
| HC1-rim | 10/6/12 | -- | -1.0 | 1.4 | 1.6 | -- | 2013 | 1.0298 | 22.0 ± 4.1 |
| HC1-trough | 8/8/12 | -- | -- | -- | -- | -- | 2017 | 1.0420 | 33.6 ± 3.5 |
| | 8/11/12 | 32 | -- | 6.3 ± 0.6 | 4.6 ± 0.2 | -- | 2017 | 1.0501 | 41.6 ± 4.1 |
| | 10/6/12 | -- | -1.0 | 1.4 | 1.5 | -- | 2013 | 0.9936 | -14.0 ± 3.2 |
| | 9/6/13 | 36 | 0.5 | 0.85 ± 0.07 | 0.85 ± 0.16 (n=3) | 1.53 ± 0.016 | 2013 | 1.0271 | 19.3 ± 3.6 |
| HC2-center | 9/7/14 | 23 | 4.0 | 3.4 ± 0.1 | 2.3 ± 0. | 0.425 ± 0.0038 | 2016 | 0.9775 | -31.3 ± 5.0 |
| HC3-center | 8/13/12 | 29 | 9.7 | 11.7 ± 1.1 (n=4) | 9.8 ± 1.6 (n=4) | -- | 2013 | 1.0408 | 32.9 ± 5.2 |
| | 9/7/14 | 31 | 4.0 | 2.8 ± 0.2 | 2.4 ± 0.4 | 0.408 ± 0.0043 | 2017 | 0.9991 | -9.0 ± 1.9 |
| HC3-trough | 7/12/13 | -- | 10 | -- | -- | 1.65 ± 0.0070 | 2013 | 1.0520 | 44.0 ± 6.0 |
| | 9/6/13 | 32 | 4.0 | -- | 3.1 ± 0.2 (n=3) | 0.489 ± 0.0033 | 2013 | 1.0211 | 13.3 ± 3.7 |
| FC1-center | 9/2/14 | 37 | 5.3 | 4.3 ± 1.5 | 2.9 ± 0.1 | 0.572 ± 0.0066 | 2016 | 1.0044 | -3.6 ± 2.9 |
| FC2-rim | 8/12/12 | 29 | -- | 4.7 ± 0.4 | 3.9 ± 0.2 | -- | 2017 | 1.0343 | 25.9 ± 2.5 |
| | 10/6/12 | -- | -1.0 | -- | -- | -- | 2013 | 1.0375 | 29.7 ± 3.3 |
| FC2-trough | 8/11/12 | 27 | -- | 3.7 ± 0.5 | 2.8 ± 0.2 | -- | 2017 | 1.0428 | 34.3 ± 3.4 |
| | 7/14/13 | 25 | 9.8 | 3.3 ± 0.1 (n=3) | 2.3 ± 0.2 (n=3) | 2.47 ± 0.011 | 2013 | 1.0372 | 29.3 ± 3.7 |
| | 9/6/13 | 30 | 3.2 | 2.1 ± 1.6 (n=3) | 1.5 ± 0.4 (n=3) | 1.50 ± 0.0076 | 2013 | 1.0171 | 9.4 ± 3.6 |
| FC3-center | 9/2/14 | 44 | 4.5 | 3.5 ± 0.2 | 2.8 ± 0.1 | 0.464 ± 0.0043 | 2016 | 1.0173 | 9.2 ± 3.6 |
| FC4-center | 9/7/14 | 34 | 2.0 | 2.4 ± 0.8 | 1.8 ± 0.7 | 0.312 ± 0.0032 | 2016 | 1.0418 | 33.5 ± 4.1 |
| FC4-trough | 9/5/13 | 36 | 5.2 | -- | 4.4 ± 0.6 (n=3) | 0.471 ± 0.0029 | 2013 | 1.0163 | 8.6 ± 3.9 |
| LC1-center | 8/10/12 | 31 | -- | 5.7 ± 0.07 | 4.3 ± 0.07 | -- | 2017 | 1.0317 | 23.3 ± 2.5 |
| | 10/6/12 | -- | -1.0 | 1.9 | 1.9 | -- | 2013 | 0.8470 | -159.5 ± 13.1 |
| | 9/7/13 | 34 | 3.2 | -- | 2.1 ± 0.2 (n=3) | 0.825 ± 0.0092 | 2013 | 1.0026 | -5.0 ± 4.4 |
| LC1-rim | 10/6/12 | -- | -1.0 | 1.8 | 1.9 | -- | 2013 | 1.0663 | 58.2 ± 7.3 |
| LC1-trough | 8/10/12 | 33 | -- | 5.5 ± 2.3 | 4.1 ± 1.5 | -- | 2017 | 1.0464 | 40.0 ± 3.7 |
| | 10/6/12 | -- | -1.0 | 1.9 | 2.1 | -- | 2013 | 1.0339 | 26.1 ± 6.5 |
| | 9/7/13 | 41 | 3.2 | -- | 2.7 ± 0.3 (n=3) | 0.320 ± 0.0033 | 2013 | 1.0389 | 31.0 ± 4.6 |
| LC2-center | 9/2/14 | 29 | 4.8 | 5.2 ± 0.1 | 4.3 ± 0.1 | 0.372 ± 0.0032 | 2016 | 1.0221 | 14.0 ± 3.8 |
| LC3-center | 10/6/12 | -- | -1.0 | 1.7 ± 0.2 | 1.6 ± 0.07 | -- | 2013 | 1.0304 | 22.5 ± 3.4 |
| | 7/12/13 | 23 | 8.2 | 12.9 ± 0.04 | 8.8 ± 0.2 | 0.607 ± 0.0032 | 2013 | 1.0336 | 25.8 ± 3.6 |
| | 9/7/13 | 33 | 2.0 | -- | 2.5 ± 0.02 (n=3) | 1.12 ± 0.019 | 2013 | 1.0386 | 30.7 ± 4.2 |
| | 9/2/14 | 27 | 5.0 | 5.1 ± 0.1 | 4.3 ± 0.0 | 0.521 ± 0.0023 | 2016 | 1.0167 | 8.6 ± 3.7 |
| LC3-trough | 10/6/12 | -- | -1.0 | -- | -- | -- | 2013 | 1.0686 | 60.5 ± 9.8 |
| | 7/12/13 | 31 | 8.2 | 10.5 ± 0.4 | 6.6 ± 0.08 | 0.674 ± 0.0021 | 2013 | 1.0445 | 36.5 ± 4.1 |
| | 9/7/13 | 37 | 2.0 | -- | 2.3 ± 0.2 (n=3) | 1.31 ± 0.0046 | 2013 | 1.0686 | 27.3 ± 4.1 |
| LC4-center | 9/7/14 | 28 | 3.1 | 7.1 ± 0.3 | 6.1 ± 0.9 | 0.463 ± 0.0029 | 2016 | 1.0031 | -4.9 ± 3.8 |

Note: -- indicates measurements for which data are unavailable.

5 *Error ranges represent standard deviations across measurements with n=2 unless otherwise noted. When only one measurement was made, no error range is reported.

**Error ranges represent standard error of the regression used to calculate CO$_2$ flux.

***Values have been corrected to exclude atmospheric CO$_2$ using a 2-pool isotopic mixing model. Error ranges represent standard errors from the 2-pool mixing model, derived from analytical error and standard deviations in the source isotopic compositions.



**Table 2. Isotopic composition of soil profile $CO_2$**

| Profile | Month | Depth (cm) | $\delta^{13}C_{CO2}$ (‰) | Analysis year | $F^{14}C$ | $\Delta^{14}C_{CO2}$ (‰) | Mean age of respired C (y) |
|---|---|---|---|---|---|---|---|
| HC1-center | 8/2012 | 31 | -- | 2013 | 0.7464 | -259.2 ± 7.5 | 3050 |
| | 9/2013 | 20 | -23.7 | 2013 | 0.7270 | -278.6 ± 2.6 | 3350 |
| | 8/2012 | 10 | -20.1 | 2013 | 0.9473 | -59.9 ± 3.0 | 770 |
| | 8/2012 | 20 | -- | 2013 | 0.8864 | -120.3 ± 3.2 | 1320 |
| HC3-center | 8/2012 | 29 | -21.7 | 2013 | 0.8944 | -112.4 ± 2.7 | 1240 |
| | 7/2013 | 10 | -- | 2013 | 0.9037 | -101.3 ± 4.4 | 1150 |
| | 7/2013 | 20 | -- | 2013 | 0.7382 | -267.4 ± 2.6 | 3170 |
| HC3-trough | 7/2013 | 10 | -- | 2013 | 0.7898 | -216.2 ± 3.3 | 2440 |
| | 7/2013 | 20 | -23.7 | 2013 | 0.8447 | -161.7 ± 2.4 | 1760 |
| FC2-center | 8/2012 | 10 | -24.8 | 2013 | 0.9509 | -56.3 ± 2.8 | 740 |
| | 8/2012 | 20 | -24.9 | 2013 | 0.9729 | -34.5 ± 2.9 | 580 |
| FC4-center | 7/2013 | 20 | -24.8 | 2013 | 0.8034 | -202.7 ± 2.1 | 2260 |
| LC3-trough | 7/2013 | 10 | -24.7 | 2013 | 1.0005 | -7.1 ± 3.9 | 410 |
| | 7/2013 | 20 | -- | 2013 | 0.8755 | -131.2 ± 3.7 | 1430 |



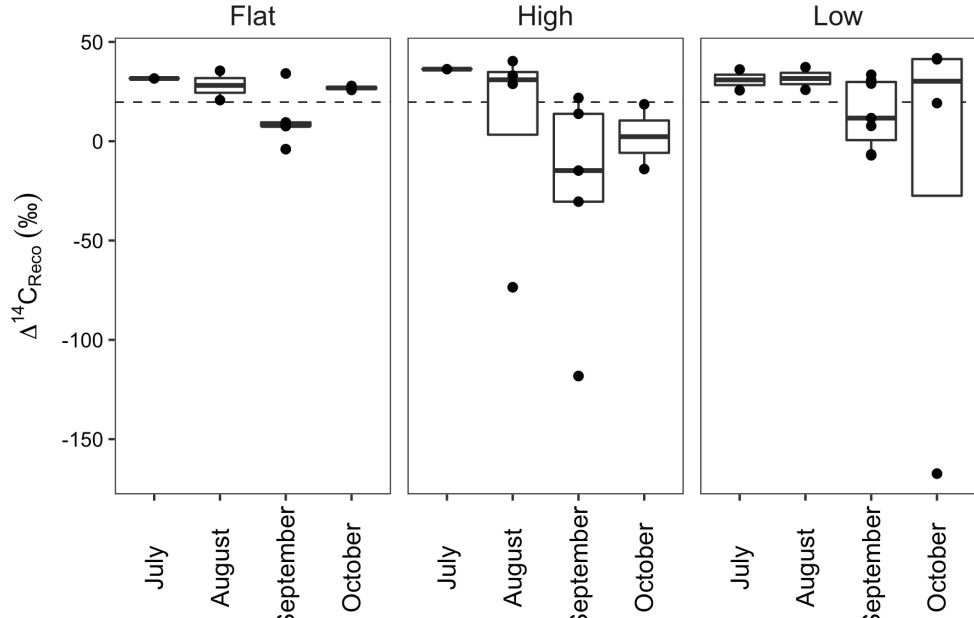

5    **Figure 1. Radiocarbon content of ecosystem respiration, separated by polygon type and sampling month. Boxes represent the first and third quartiles, and whiskers extend to the farthest values within 1.5 times this range. Dashed horizontal line indicates the mean $\Delta^{14}$C-$CO_2$ of the local atmosphere during the 2012-2014 summer seasons.**





5    **Figure 2. Ecosystem respiration rate (a) and radiocarbon content (b), measured from polygon centers in September 2014. Error bars represent standard error with n=3. Dashed horizontal line indicates $\Delta^{14}C$-$CO_2$ of the local atmosphere at the time of sampling.**



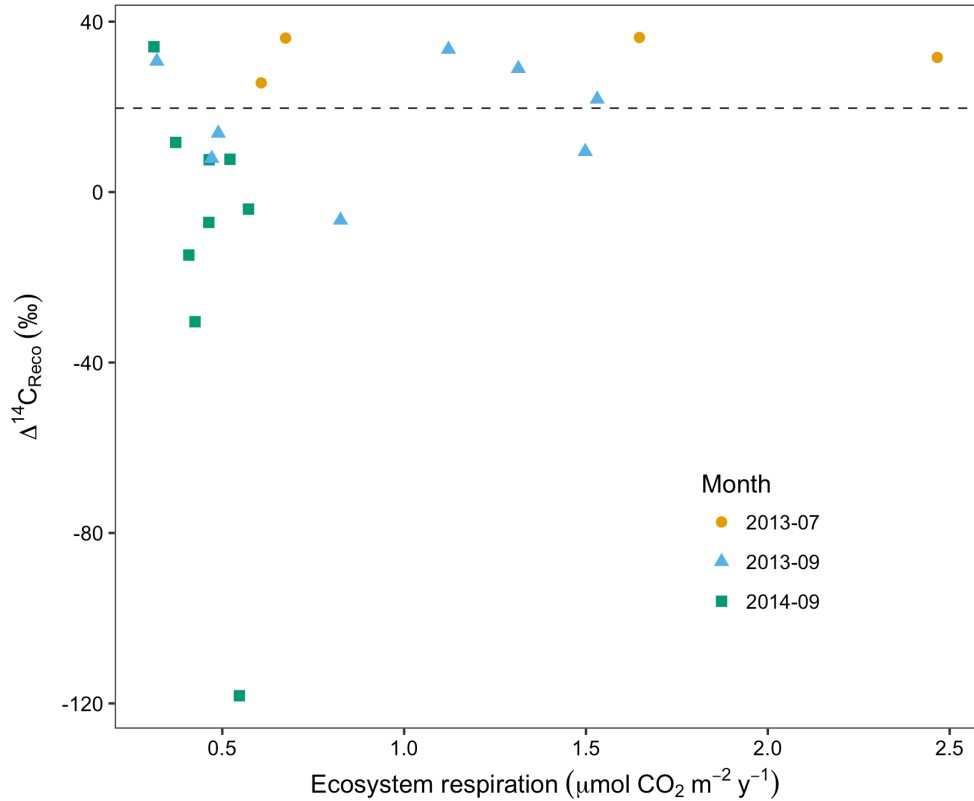

5    **Figure 3. Rates and radiocarbon contents of ecosystem respiration in July and September 2013 and September 2014.  Dashed horizontal line indicates the mean $\Delta^{14}$C-CO$_2$ of the local atmosphere during the 2013 and 2014 summer seasons.**





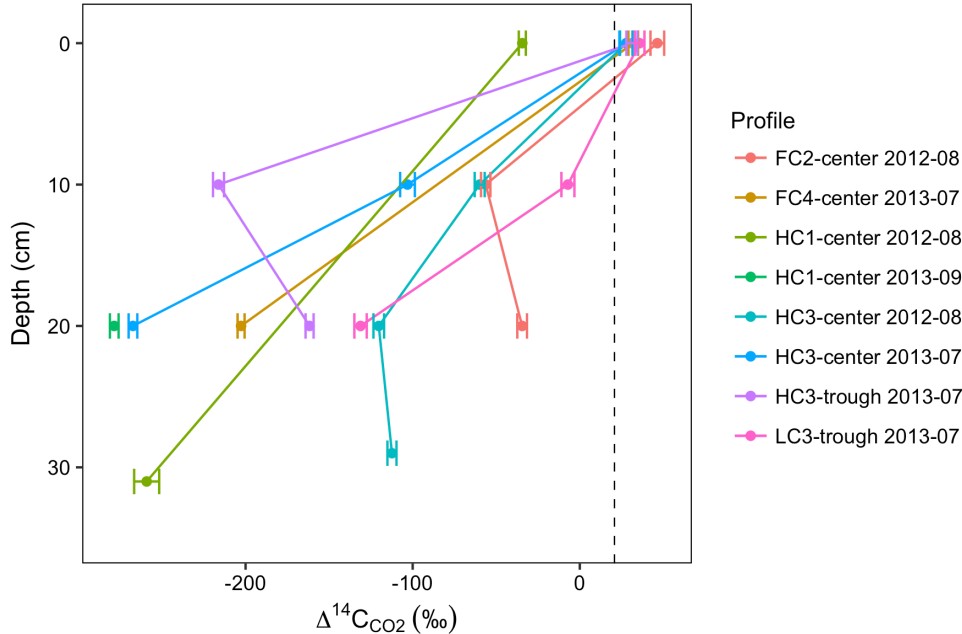

**Figure 4. Soil depth profiles of $\Delta^{14}$C-CO$_2$.** Lines connect samples collected in the same vertical profile and month, and error bars represent analytical error. Samples from 0 cm depth were collected from surface soil chambers and reflect CO$_2$ that accumulated

10     after chambers were scrubbed of CO$_2$. All other samples were collected from soil wells. Dashed vertical line indicates the mean $\Delta^{14}$C-CO$_2$ of the local atmosphere during the 2012 and 2013 summer seasons.