# Peer review of "Radiocarbon Measurements of Ecosystem Respiration and Soil Pore-Space CO2 in Utqiagvik (Barrow), Alaska"

_Earth System Science Data, 2018_

## Referee Comment (RC1) · Anonymous Referee #1 · 14 Jun 2018

General comments: Overall quality & discussion of the paper. This manuscript presents a unique and important dataset of  14C-CO2 soil and ecosystem respiration from a high-latitude polygonal tundra site. The data are important because they provide insight to the stability of old permafrost C and the conditions under which old permafrost C stocks could be released to the atmosphere. Overall the paper is well written although I think that it would be helpful to have more background about the ecosystem and context for why the different polygonal tundra features might affect  14C-CO2. Specifically, what might explain some of the  14C differences shown in Figure 1 and 2? If thaw depth does not explain the differences, is there anything else that might? Moisture, vegetation type, organic soil content, amount of cryotrubation?

The respired and profile  14C measurements are a very nice complement to each other because they really let us understand production vs. release mechanisms for soil C. One challenge with the profile data is that we don't know the time scale over which the old 14C has accumulated. There could be lots of really old 14C because of slow diffusion rates and high accumulation. For example (Lee et al. 2010) measure very high Co2 concentrations deep in the soil profile and attribute this to low diffusion rates rather than high production rates. Of course this is difficult to solve and perhaps under steady-state assumption the accumulated CO2 is constantly being produced and diffusing out of the soil profile, and the 14C reflects the decomposability of old C and therefore it's eventual release to the atmosphere. Perhaps the authors could add 1-2 sentences about this, simply to point out some of the complexities with interpreting the data. A few additional comments below point out a few places where more consistent data presentation would make the manuscript more reader friendly and reduce some confusion that I encountered. Beyond that, I think this manuscript documents an important and interesting data set and should be published.

Data and code could be accessed with the DOI and links provided!

Specific comments: Individual questions & issues, and technical comments

Overall data presentation: Year/month is inconsistent. Sometimes month is reported with year, sometimes without. For example, figure 1 ignores years, while figure 4 explicitly represents years. That's confusing. How important do the authors think that year is? Can year be left out?

Table & Figure comments: Can a seasonal Reco flux rate figure be added?

Tables: The tables are tough to read. Could some of the environmental data be summarized in a figure and the tables moved to the supplement? As I understand it all the CO2 flux data is shown in figures so the tables aren't critical for the reader to understand the patterns.

Figure 2: Is the data the same as in Figure 1, September? It looks different. ..... Flat has 14C<0 in Figure 1 and >0 in Figure 2

Figure 3: Can the month in the legend be written as a month name (ie: July, September)? That would be much easier to read.

Figure 4: Can the legend be Flat2-Center-August 2012, Flat4 Center July 2013, etc? Would be easier to read. Even if there is no overall temporal and spatial pattern could the lines in the figure be systematically grouped? One colour for each location, and a different symbol+line type for early/late months? It might be conceptually helpful to have a horizontal line at 0cm to indicate the soil surface, and perhaps put the chamber flux data at +2cm?

Line-by-line minor comments: Page 1: Line 29-30: Cite Bond-Lamberty soil respiration database paper?

Line 34: something is missing in the end of the sentence, the grammar/tense is wrong: 'heterotrophic decomposition of soil carbon that cycles on broad range of timescales'

Page 2: Line 5: 'thaw depth' is not an obvious variable here without introducing permafrost? To some extent thaw depth is captured by soil temperature. Perhaps 'soil C pool' would be useful to add? Or maybe 'permafrost state'?

Line 6: what does 'such variations' refer to?

Line 7: the jump from environmental controls to use or availability of soil C substrate pools is a bit unclear. I think clarifying whether 'such variations' refers to 14C or variation in environmental factors would help. I suggest explicitly naming the variation that is meant, rather than 'such variations'.

Line 11: parameters of what?

Line 17: consider also citing (Elberling et al. 2013, Schädel et al. 2013)

Line 21-22: something is missing from this sentence

Line 24: (Schuur et al. 2009, Nowinski et al. 2010) ?

Line 26: It would be helpful to explain, in a few sentences, what polygonal tundra is, why it's important, and what unique features it has (eg: drained vs saturated microsites, C accumulation, temperature regimes). (Ping et al. 2015) might be a useful reference. This is mentioned in the methods, and I think it would be worth a brief mention in the introduction too.

Page 3:

Line 17: chamber height? Or volume?

Line 19-20: Oh, I see. I would move this sentence one earlier.

Page 5: Line 3: what is the mean 13C value of these samples? Is it possible that this 13C value largely represents autotrophic respiration, rather than soil respiration? My guess would be that the chambers with rapid CO2 accumulation and the highest Co2 concentrations have high plant respiration.

Line 10: This is a good idea for dual filtering criteria. I like it.

Page 7:

Line 25: is the data in figure 2 a subset of figure 1? The patterns between polygons in September look different in the two figures, and I can't understand why.

Line 30: profiles of what?

Line 32: I feel this needs a little more elaboration: 'At the scale of individual profiles seasonal variations in  14C Reco correspond with changes in thaw', that's inferred from seasonal pattern of  14C Reco decreasing as thaw exposes deeper parts of the soil profile? In contrast, across sites, there is no correlation between thaw and  14C Reco.

Page 8: Line 6: I think this should be reworded to something like: 'As a result, old,

slow-cycling C from deep Reco comprises a large percentage of the total C flux only when autototrophic and surface soil (or fast-cycling) contributions are low. I think that might be a more accurate generalization, rather than old soil contributions being high when Reco rates are low, because there could be a number of reasons for low Reco rates like overall low plantµbial activity, which might not affect the  14C.

Line 18 -24: That's really interesting! Line 19: Should this be 'Figure 4'??

Line 24: The reason why cryoturbation may explain the more positive  14C at depth may only be obvious to people familiar with permafrost dynamics? One sentence would be sufficient to say that cryoturbation can transport large chunks of surface/organic material deeper into the profile.

Page 10: Line 5: But these slow cycling contributions might be missed when measuring surface  14C fluxes alone?

Line 14: the distinction between newly thaw and historical annual thaw might be very difficult for people without an Arctic/permafrost background to understand. Perhaps elaborate a little what this means and why it matters. Newly thawed does not refer to new C, it is newly exposed old C, I think that's a very permafrost-specific concept.

References

Elberling, B., A. Michelsen, C. Schädel, E. A. G. Schuur, H. H. Christiansen, L. Berg, M. P. Tamstorf, and C. Sigsgaard. 2013. Long-term CO2 production following permafrost thaw. Nature Climate Change 3:890–894.

Lee, H., E. A. G. Schuur, and J. G. Vogel. 2010. Soil Co2 Production in Upland Tundra Where Permafrost Is Thawing. Journal of Geophysical Research 115:1–11.

Nowinski, N. S., L. Taneva, S. E. Trumbore, and J. M. Welker. 2010. Decomposition of old organic matter as a result of deeper active layers in a snow depth manipulation experiment. Oecologia 163:785–792. Ping, C. L., J. D. Jastrow, M. T. Jorgenson, G. J. Michaelson, and Y. L. Shur. 2015. Permafrost soils and carbon cycling. SOIL 1:147–

171.

Schädel, C., E. A. G. Schuur, R. Bracho, B. Elberling, C. Knoblauch, H. Lee, Y. Luo, G. R. Shaver, and M. R. Turetsky. 2013. Circumpolar assessment of permafrost C quality and its vulnerability over time using long‐term incubation data. Global Change Biology 20:641–652.

Schuur, E. A. G., J. G. Vogel, K. G. Crummer, H. Lee, J. O. Sickman, and T. E. Osterkamp. 2009. The effect of permafrost thaw on old carbon release and net carbon exchange from tundra. Nature 459:556–559.

---

## Referee Comment (RC2) · Anonymous Referee #2 · 18 Jun 2018

General comments: The authors present soil respiration quantity and 14CO2 signature as well as soil pore 14CO2 data of high-latitude soils during the thawing seasons of 2012 -2014. Acquiring this type of data in these regions is not trivial and the data provides some interesting insights into the seasonally and spatially variable carbon respiration in this region. The data is provided under the given link and the R codes used for the graphs and modeling are provided on github. Overall, the data and presented insights are valuable and merit publication.

Some aspects regarding the sampling documentation, blank analyses, statistics as well as data use in models could be improved upon in order for this dataset in order

to reach its full potential. Specifically, as micro topographic variability was identified as a key driver for variability in 14CO2 trends, a visualization and quantification of that topography would be helpful. Furthermore, respiration data of soil carbon is directly linked to the solid bulk soil carbon – however, that there is no data on this pool. Although analytical precision data was provided, no data on procedural blanks were provided. This type of sampling and measurements take many intermediate steps, and if available it would be good to have procedural blank data in addition to just the OXAs (e.g. Hanke et al., 2017). Especially because the samples were processed not in one batch but rather over a number of years (2014-2017). Regarding statistics, there are some points were statistical terms such as standard deviation is used when n=2, or box plots with quartiles are implemented when n = 3 or 4. This is could be seen as misleading and result in misinterpretation. See specific comments for suggestions how this can be improved. Considering that this is a journal aimed at geo-data and use of this data in other studies, the authors could add details regarding which specific types of models could benefit from this data. This could improve how this data is used in the future stages.

Specific comments: Regarding the novelty of the data and methods, the data is novel and focuses on a region which is undergoing rapid environmental change. Data is based on established sampling and analytical methods.

Regarding the data being used in the future: The authors refer to the data being used in 'models', but could perhaps provide more detail. At some point, they refer to a publication (He et al., 2016) that uses Earth System models, but I can imagine that there could be a larger scope of the implementation of this data in models. One major piece of information which is missing - which is likely important for these types of models - is the size and 14C signature of the solid soil carbon. It is unknown how the amount of respired carbon relates to this –presumably much larger – pool.

One additional aspect which can be improved upon in order to increase the value of this data is an improved description of the morphology of the micro topographic features

that were sampled. The authors underline that micro topography has a strong impact on 14C signature of respired carbon, but there are no pictures or sketches. I would recommend adding photographs or a sketch showing the features. Furthermore, in order to be able to extrapolate this data, it would be helpful to know how often which type of polygon types occurs. That way, the signal could be averaged for this region.

The materials and methods are described in sufficient detail and the R code and data themselves are provided. The article itself is appropriate for the publication of this dataset because it clearly describes how the samples was acquired and data was measured. As mentioned previously, if there is data on the bulk carbon quantity and signature from other papers, it could be valuable to cite and integrate that. Presently it is not present in the paper.

Data Quality & accessibility: The data is accessible & codes are available on git. There are even some 2-pool model codes on git which I believe are not used in the paper. Error estimates could be improved upon. Mostly only the analytical error is provided. I am missing the procedural blank, which is especially important as the samples were measured over three years. The sample sizes for 14C measurements are also not provided. Smaller samples are more susceptible to contamination (Hanke et al., 2017). Ideally, there would be more sample replicates. However, considering the difficulty of the sampling and the significant cost of 14C measurements it makes sense that no extensive replication done in this case.

There is also a potential bias in the soil respiration data that has not yet been addressed. The authors state that when the soil was water saturated, it was not possible to measure pore CO2. It is very likely that there is a difference in carbon decomposition speed between water saturated and non-water saturated soils. Therefore, if only the non-saturated soils soil pore CO2 is measured, there could be a bias in the interpretation. Authors should address this in the discussion.

Dataset quality. Data seems in the range of what can be expected from these types

of cold soils. Dataset presentation: The graphs can be improved upon on a number of points, especially concerning the usage of some statistical terms, as detailed in the section technical comments. Note that box plots are not designed for data where n = 3. Nor is standard deviation applicable if n = 2. Standard deviation is a measure to see what the spread of data is, but with n= 2 you don't really have a spread. Publication length: The length of the paper is good, the figures and table are appropriate. If the authors provide more information of the sampling and micro topography, the dataset could be used in the future after reading the paper & downloading the data

Technical comments are attached in PDF format

References Graven HD (2015) Impact of fossil fuel emissions on atmospheric radio-carbon and various applications of radiocarbon over this century. Proceedings of the National Academy of Sciences, 1–4. Hanke UM, Wacker L, Haghipour N, Schmidt MWI, Eglinton TI, McIntyre CP (2017) Comprehensive radiocarbon analysis of ben-zene polycarboxylic acids (BPCAs) derived from pyrogenic carbon in environmental samples. Radiocarbon, 59, 1103–1116.

Please also note the supplement to this comment:
https://www.earth-syst-sci-data-discuss.net/essd-2018-29/essd-2018-29-RC2-supplement.pdf
* * *
[Figure]
Technical comments

| Page | Line | Comment |
|---|---|---|
| 1 | 11 | Specify model |
| | 12 | Specify use of D14C |
| | 25 | You state the flux is critical – is there any quantification? |
| 2 | 12-15 | Sentence is very long & not clear. |
| | Study site section | If possible, add a sketch of polygon and add how dominant each polygon type is |
| 4 | Surface emissions | Inserting the chambers into the soil may disturb the soil and increase gas exchange with the deeper soil: this should be addressed in the discussion |
| 5 | 2-3 | 'measurements lacking a linear range were not included in the dataset' Do you mean samples that fall outside of the linear range, or complete measurement series? |
| | 16 | Pore $CO_2$ was not measurement for water saturated soils. This may bias final interpretations. Please address |
| | 14 | Low-concentrations pore $CO_2$ measurements were omitted except when value was clearly not-atmospheric. Does that mean that the excluded samples all had an atmospheric signature? |
| 6 | 11 | Tr or mean the of carbon in plants is either assumed to be 0 or 5 year. Whilst cited sources give a which wider range. Even decomposing pine needles in a temperate zone are generally 8 years old. In such cold environments with slow growth rates one could expect even older ages of plant-derived carbon into the soil. Is this assumption realistic? Please add some further justification. |
| | 21-25 | When two turnover times are equally likely (Graven, 2015), it could also be helpful to have measured the bulk soil signature. If this is indeed not available, this is the best fix possible, but it is not ideal. |
| 7 | 14 | When describing the effects of the polygons and microtopography, a sketch or image would be helpful for the understanding of the reader |
| 8 | 10 | Authors state that pre [14]$CO_2$ becomes increasingly negative with depth, but this is not the case for three profiles (as mentioned later in the text on line 18 |
| | 13 | Author sate that there are minimal contributions from plant-respired carbon. Did they do a mass-balance? Or is this a qualitative statement? Please specify. |
| | 13-17 | These statements support Figure 4. For better flow, maybe move towards end of section |
| | 19 | Figure 2 should be 4 |
| | 25-26 | When trying to understand soil carbon stability and decomposability, it would be good to not only determine the age of respired $CO_2$ but also the age of the bulk soil carbon. If this is not |

**Fig. 1.**
| | | available for these profiles, but similar ones, it would be good to benchmark the data. |
|---|---|---|
| 9 | 9 | Authors state: "due to sampling limitations, soil profile 14C data are available from only a subset of polygon types […] In tables 1 & 2 I only see soil CO2 measurements, no bulk. Did they mean to refer to pore CO2? |
| | 35 | 'later in season' please be more specific |
| Tables | 1 | Standard deviation with n = 2 does is not appropriate and can be seen as misleading (go back to calculation). It is supposed to give the range of sample variability, but for n =2 that is not applicable. Consider an alternative, e.g. giving the average |
| | 2 | HC1-center: Measurements were done at different depths during different years. Or is it a typo? If not, considering the inter annual variability, is it reasonable to compare the data? Please clarify.

 For HC3 center for the same depth there are large differences in 14C |
| Figures | 1 | It appears that boxplot R are used for n =2-5. It may not be the most appropriate way of presenting the range of data. Also add in the explanation that when R boxplots are visualized, data that falls outside the range is statistically speaking an outlier. Also indicate what the solid line in the box means. Also considering adding a striped line for the mean

 Considering using this plot to highlight the important finding that during high-summer, all respiration is from year to decadal old C, whilst later in the season, whilst the deeper soil continues to warm, older (stabilized) C is lost. |
| | 2 | Consider changing axis (starting at 0.3) in a to highlight differences in respiration |
| | 3 | Legend should be month-year
 This graph shows an interesting trend which has not been explicitly discussed in the text, but which could be valuable for the data interpretation. There could be two end member type of behaviours: High summer (2013-07), high respiration of topsoil C releases bomb-enriched carbon. Late summer (2014-09) releases a low amount of old, stabilized carbon. 2013-09 could be an intermediate type where there is mixing from both pools, providing a spread of ages and concetrations. |
| | 4 | Colour spread is not optimal, some colours are near-identical. Considering changing. Also consider highlighting the 3 soils which some younger respired C at depth, or grouping samples by polygon type. |

**Fig. 2.**

---

## Author Comment (AC1) · 26 Jul 2018

**Response to anonymous Referee #1**

We welcome this reviewer's thoughtful comments and suggestions and have provided responses to individual comments below. Our line-by-line responses describe changes that will be submitted in the revised manuscript.

General comments

*This manuscript presents a unique and important dataset of ï¿A˙D 14C-CO2 soil and ecosystem respiration from a high-latitude polygonal tundra site. The data are important because old permafrost C stocks could be released to the atmosphere. Overall the paper is well written although I think that it would be helpful to have more background about the ecosystem and context for why the different polygonal tundra features might affect ï¿A˙D 14C-CO2. Specifically, what might explain some of the ï¿A˙D 14C differences shown in Figure 1 and 2? If thaw depth does not explain the differences, is there anything else that might? Moisture, vegetation type, organic soil content, amount of cryotrubation?*

At the reviewer's request, we have added a paragraph to the introduction providing further background on this ecosystem and its polygonal tundra features. In particular, we have added a brief discussion of how microtopography, hydrology, vegetation, and cryoturbation might influence carbon cycling rates in polygon tundra. In addition to this introductory material, we also added text to the discussion regarding other ecological properties that may explain the observed $\Delta^{14}$C differences:

"At the scale of an individual soil profile, seasonal variations in $\Delta^{14}$C$_{Reco}$ correspond with changes in thaw depth. At the site scale, however, thaw depth may not be a useful predictor of spatial variations in $\Delta^{14}$C$_{Reco}$. Instead, the spatial distribution of carbon cycling rates may be more directly influenced by profile-specific properties that do not necessarily correlate with thaw depth. These factors—such as organic layer thickness, vegetation composition, productivity and rooting depth, the presence of cryoturbation, and oxygen availability to decomposers—vary according to polygon morphology (Newman et al., 2015; Ping et al., 2015; Sloan et al., 2014; Vaughn et al., 2016). Accordingly, these properties may underlie the differences in $\Delta^{14}$C$_{Reco}$ we observed in September 2014 between HC polygons and the other polygon types (Fig. 2)."

*The respired and profile ï¿A˙D 14C measurements are a very nice complement to each other because they really let us understand production vs. release mechanisms for soil C. One challenge with the profile data is that we don't know the time scale over which the old 14C has accumulated. There could be lots of really old 14C because of slow diffusion rates and high accumulation. For example (Lee et al., 2010) measure very high Co2 concentrations deep in the soil profile and attribute this to low diffusion rates rather than high production rates. Of course this is difficult to solve and perhaps under steady-state assumption the accumulated CO2 is constantly being produced and diffusing out of the soil profile, and the 14C reflects the decomposability of old C and therefore it's eventual release to the atmosphere. Perhaps the authors could add 1-2 sentences about this,*

*simply to point out some of the complexities with interpreting the data.*

We agree with the reviewer that profile data are challenging to interpret because of unknown diffusion and accumulation rates and vertical mixing between depths. Because of these challenges, rates of old carbon release in soil surface emissions do not directly indicate when and how rapidly old carbon at depth is decomposing, nor do the depths of pore-space measurements necessarily correspond to the depths of $CO_2$ production. When interpreting soil pore-space data and linking it to co-located surface fluxes, it is important to consider both of these issues. Given the scope of our data, we were careful not to infer absolute rates or specific depths of old $CO_2$ production, but agree that it is worth addressing this point more clearly. We have added two passages discussing aspects of this issue in the manuscript's discussion:

"Detecting and characterizing the decomposition of older, deeper soil organic carbon requires direct measurements of soil pore-space $CO_2$. Such measurements provide a qualitative indicator of old carbon decomposition; as with surface $CO_2$ effluxes, proportional or absolute contributions from distinct carbon pools cannot be calculated without well-resolved vertical distributions of $^{14}C$ and $^{13}C$ source pools, as well as gas transport within the profile."

"Because of unknown diffusion rates within the soil profile, it is challenging to quantitatively use pore-space $CO_2$ measurements to link soil carbon cycling rates to soil surface $\Delta^{14}C_{Reco}$. Low diffusion rates can lead to the accumulation of high concentrations of $^{14}C$-depleted, slow-cycling $CO_2$ deep in the soil profile (Lee et al., 2010), and vertical mixing can transport $CO_2$ away from the site of production prior to its collection as soil pore gas."

*A few additional comments below point out a few places where more consistent data presentation would make the manuscript more reader friendly and reduce some confusion that I encountered. Beyond that, I think this manuscript documents an important and interesting data set and should be published.*

*Data and code could be accessed with the DOI and links provided!*

Specific comments

Overall data presentation

*Year/month is inconsistent. Sometimes month is reported with year, sometimes without. For example, figure 1 ignores years, while figure 4 explicitly represents years. That's confusing. How important do the authors think that year is? Can year be left out?*

Our choice to include year in the Figure 4 legend was not intended to emphasize the importance of year, but rather to clearly define each radiocarbon depth profile displayed in the figure. Based on this comment and the reviewer's specific suggestions below, we have modified Figure 4 substantially, now grouping soil profiles by polygon type and position. With this new formatting, year and specific core name are no longer included in

the legend.  Additionally, we have modified Figure 1 substantially, based on comments from Reviewer #2.  The new version includes day of year on the x-axis in place of sampling month.  We have also edited the caption to clarify the fact that data in Figure 1 were compiled across the 3 sampling years.

Table and Figure comments

*Can a seasonal Reco flux rate figure be added?*

Because Reco was measured on only a subset of sampling dates, we have chosen not to include a seasonal Reco flux rate figure.  From different soil profiles at the same site, two published studies (Vaughn et al., 2016 and Wainwright et al., 2015) both show a clear seasonal decrease in Reco rate from July/August into October.  Although we have not added a seasonal Reco figure in this manuscript, we will instead include a reference to these two datasets in the revised manuscript.  With this reference, we will briefly discuss how these seasonal Reco trends support our interpretation that slow-cycling carbon sources contribute large proportions of total ecosystem respiration only during times of low overall Reco.

*Tables: The tables are tough to read. Could some of the environmental data be summarized in a figure and the tables moved to the supplement? As I understand it all the CO2 flux data is shown in figures so the tables aren't critical for the reader to understand the patterns.*

Based on this suggestion, we will move Table 1 to the supplement and add two additional figures: (1) soil temperature by month, and (2) a third panel in Figure 2 showing thaw depth in September 2014.  We chose to leave Table 2 in the main document because it shows the results of the mean age of respired carbon calculation, and we felt this table was not as unwieldy as Table 1.

*Figure 2: Is the data the same as in Figure 1, September? It looks different: : :.. Flat has 14C<0 in Figure 1 and >0 in Figure 2*

Yes, these figures represent different data.  Figure 1 includes all sampling years (which for September, includes 2013 and 2014), whereas figure 2 includes only September 2014, which we highlight as a balanced sub-dataset that clearly demonstrates the influence of microtopography on $\Delta^{14}C_{Reco}$.  We have clarified this in the Figure 1 caption, and by adding the following sentences to the main document: "In September 2014, we measured $\Delta^{14}C_{Reco}$ and ecosystem respiration rates from the centers of three polygons of each type (Fig. 2).  The influence of microtopography on old carbon emissions is particularly apparent in this complete and evenly distributed measurement set."

*Figure 3: Can the month in the legend be written as a month name (ie: July, September)? That would be much easier to read.*

The figure legend has been changed as suggested.

*Figure 4: Can the legend be Flat2-Center-August 2012, Flat4 Center July 2013, etc? Would be easier to read. Even if there is no overall temporal and spatial pattern could the lines in the figure be systematically grouped? One colour for each location, and a different symbol+line type for early/late months? It might be conceptually helpful to have a horizontal line at 0cm to indicate the soil surface, and perhaps put the chamber flux data at +2cm?*

Based on this suggestion and the comment above, we have reformatted Figure 4.  Figure 4 now groups $\Delta^{14}C_{CO2p}$ profiles by position within polygon and polygon type and no longer lists the specific profile ID and measurement month/year.  We have also added a horizontal line at 0 representing the soil surface and moved the chamber flux data to +2 cm depth.

Line-by-line minor comments

*Page 1: Line 29-30: Cite Bond-Lamberty soil respiration database paper?*

We have added this reference.

*Line 34: something is missing in the end of the sentence, the grammar/tense is wrong: 'heterotrophic decomposition of soil carbon that cycles on broad range of timescales'*

To improve readability, we have changed this sentence to the following: "$CO_2$ emitted from the soil surface includes autotrophic respiration of rapidly cycling carbon as well as heterotrophic decomposition of carbon that cycles on broad range of timescales."

*Page 2: Line 5: 'thaw depth' is not an obvious variable here without introducing permafrost?  To some extent thaw depth is captured by soil temperature. Perhaps 'soil C pool' would be useful to add? Or maybe 'permafrost state'?*

Following the reviewer's suggestion, we have removed the words "thaw depth" and explicitly mentioned carbon pools earlier in this sentence.

*Line 6: what does 'such variations' refer to?*

For clarity, we have changed the sentence to read, "variations in the radiocarbon abundance of respired $CO_2$…"

*Line 7: the jump from environmental controls to use or availability of soil C substrate pools is a bit unclear. I think clarifying whether 'such variations' refers to 14C or variation in environmental factors would help. I suggest explicitly naming the variation that is meant, rather than 'such variations'.*

We believe that the added mention of carbon pools in line 5 and language clarification in line 6 have addressed this comment.

*Line 11: parameters of what?*

We have changed this to "carbon pool-specific respiration rate parameters."

*Line 17: consider also citing* (Elberling et al., 2013; Schädel et al., 2014)

We have added a reference to Elberling et al., 2013.

*Line 21-22: something is missing from this sentence*

To improve readability, we have changed this sentence to read, "to quantify in situ decomposition rates, field radiocarbon measurements can be used to differentiate between slow-cycling and fast-cycling carbon and link decomposition dynamics to environmental controls.

*Line 24: (Nowinski et al., 2010; Schuur et al., 2009)?*

Both references have been added.

*Line 26: It would be helpful to explain, in a few sentences, what polygonal tundra is, why it's important, and what unique features it has (eg: drained vs saturated microsites, C accumulation, temperature regimes). (Ping et al., 2015) might be a useful reference. This is mentioned in the methods, and I think it would be worth a brief mention in the introduction too.*

We agree with this suggestion and have added a brief paragraph to the introduction describing polygon tundra and its relevance to carbon cycling. To the methods section, we have also added and estimate of the spatial distribution of the three polygon types across the study region.

*Page 3:*
*Line 17: chamber height? Or volume?*
*Line 19-20: Oh, I see. I would move this sentence one earlier.*

As suggested, we moved the chamber height description one sentence earlier.

*Page 5: Line 3: what is the mean 13C value of these samples? Is it possible that this 13C value largely represents autotrophic respiration, rather than soil respiration? My guess would be that the chambers with rapid CO2 accumulation and the highest Co2 concentrations have high plant respiration.*

Page 5 line 3: We have added the Reco $\delta^{13}C$ end-member values to the text (-24.6, -26.5, and -26.2 ‰ for low-centered, flat-centered, and high-centered polygons respectively). We believe the reviewer is correct that these $\delta^{13}C$ values may be largely influenced by autotrophic respiration from aboveground vegetation and roots. Because we are using this background atmosphere correction to determine $\Delta^{14}C$ of ecosystem respiration,

which includes autotrophic respiration, a strong autotrophic signal in end-member $\delta^{13}$C values should not invalidate our atmospheric contamination correction.

*Line 10: This is a good idea for dual filtering criteria. I like it.*

*Page 7:*
*Line 25: is the data in figure 2 a subset of figure 1? The patterns between polygons in September look different in the two figures, and I can't understand why.*

As discussed above, figure 1 includes 2012, 2013, and 2014, whereas figure 2 is a detail of just September 2014 data. We chose to highlight this September 2014 data subset because it was balanced and complete for both Reco and $\Delta^{14}$C, and it clearly demonstrates spatial patterns. We have added clarifying sentences in the text where Figure 2 is introduced and have clarified the Figure 1 caption.

*Line 30: profiles of what?*

We have changed this to read, "across soil profiles…"

*Line 32: I feel this needs a little more elaboration: 'At the scale of individual profiles seasonal variations in ï ¿ AˇD 14C Reco correspond with changes in thaw', that's inferred from seasonal pattern of ï ¿ AˇD 14C Reco decreasing as thaw exposes deeper parts of the soil profile? In contrast, across sites, there is no correlation between thaw and ï ¿ AˇD 14C Reco.*

Based on the reviewer's suggestion, we have reworded the end of this paragraph to clarify the distinction between patterns across time in an individual soil profile and patterns across space at a single time point. This section now reads, "Interestingly, soil thaw at this time was deepest in FC polygons; from this set of September 2014 measurements, we saw no correlation across soil profiles between thaw depth and $\Delta^{14}$C$_{Reco}$. In contrast, repeated measurements from individual soil profiles indicate that $\Delta^{14}$C$_{Reco}$ tends to decrease as thaw depth increases and exposes deeper soil layers to unfrozen conditions. These findings suggest that the relationship between the depth of thaw and old carbon mineralization depends on the spatial and temporal scales of observation. At the scale of an individual soil profile, seasonal variations in $\Delta^{14}$C$_{Reco}$ correspond with changes in thaw depth. At the site scale, however, thaw depth may not be a useful predictor of spatial variations in $\Delta^{14}$C$_{Reco}$."

*Page 8: Line 6: I think this should be reworded to something like: 'As a result, old, slow-cycling C from deep Reco comprises a large percentage of the total C flux only when autototrophic and surface soil (or fast-cycling) contributions are low. I think that might be a more accurate generalization, rather than old soil contributions being high when Reco rates are low, because there could be a number of reasons for low Reco rates like overall low plantµbial activity, which might not affect the ï ¿ AˇD 14C.*

We agree with the reviewer's consideration and have reworded this sentence to incorporate their suggestion. The changed text now reads, "As a result, old, slow-cycling carbon from deep soil respiration comprises a large percentage of the total carbon flux only when respiration rates are low from autotrophic and shallow (fast-cycling) soil sources."

*Line 18 -24: That's really interesting! Line 19: Should this be 'Figure 4'??*

Yes, this was a typo and should be Fig. 4. Good catch! This has been changed in the text.

*Line 24: The reason why cryoturbation may explain the more positive ï ̨ A ̌D 14C at depth may only be obvious to people familiar with permafrost dynamics? One sentence would be sufficient to say that cryoturbation can transport large chunks of surface/organic material deeper into the profile.*

Following the reviewer's suggestion, we have added a brief definition of cryoturbation.

*Page 10: Line 5: But these slow cycling contributions might be missed when measuring surface ï ̨ A ̌D 14C fluxes alone?*

This is a good point that was not sufficiently expressed in our original manuscript. We have restructured this paragraph to emphasize that soil pore-space $CO_2$ measurements demonstrate that old, slow-cycling carbon is being decomposed, even when it does not contribute a substantial portion of the surface respiration flux. This paragraph now begins, "Measurements of radiocarbon in late-season ecosystem respiration indicate that carbon that cycles on millennial timescales contributes substantially to soil respiration. When thaw depth approached its maximum in September and October, highly depleted $^{14}C$ in respiration indicated that carbon older than 1000 years was a major source of heterotrophic respiration. Decomposition of old, slow-cycling soil carbon, however, may be missed when measuring surface $\Delta^{14}C_{Reco}$ alone. In the soil pore-space, $\Delta^{14}C_{CO2p}$ declined steeply with depth…"

*Line 14: the distinction between newly thaw and historical annual thaw might be very difficult for people without an Arctic/permafrost background to understand. Perhaps elaborate a little what this means and why it matters. Newly thawed does not refer to new C, it is newly exposed old C, I think that's a very permafrost-specific concept.*

We have added a sentence to clarify this concept for readers less familiar with Arctic soils. This section now reads, "As climate change alters these environmental controls and soils warm and thaw, a key question is how decomposition rates will change. Where permafrost degradation occurs, either through gradual deepening of the active layer or rapid thaw events, old soil carbon that has remained frozen for years to millennia is exposed to thawed conditions. A particularly important—but unknown—factor is the decomposition rate of this carbon released from thawing permafrost (Hicks Pries et al., 2013; Koven et al., 2015; Kuhry et al., 2013). Our measurements cannot differentiate

between such newly thawed soil organic matter and carbon that has historically experienced an annual thaw."

References

Elberling, B., Michelsen, A., Schädel, C., Schuur, E. A. G., Christiansen, H. H., Berg, L., Tamstorf, M. P. and Sigsgaard, C.: Long-term CO2 production following permafrost thaw, Nat. Clim. Change, 3(10), 890–894, doi:10.1038/nclimate1955, 2013.

Hicks Pries, C. E., Schuur, E. A. G. and Crummer, K. G.: Thawing permafrost increases old soil and autotrophic respiration in tundra: Partitioning ecosystem respiration using δ13C and Δ14C, Glob. Change Biol., 19(2), 649–661, doi:10.1111/gcb.12058, 2013.

Koven, C. D., Lawrence, D. M. and Riley, W. J.: Permafrost carbon−climate feedback is sensitive to deep soil carbon decomposability but not deep soil nitrogen dynamics, Proc. Natl. Acad. Sci., 201415123, doi:10.1073/pnas.1415123112, 2015.

Kuhry, P., Grosse, G., Harden, J. W., Hugelius, G., Koven, C. D., Ping, C.-L., Schirrmeister, L. and Tarnocai, C.: Characterisation of the Permafrost Carbon Pool, Permafr. Periglac. Process., 24(2), 146–155, doi:10.1002/ppp.1782, 2013.

Lee, H., Schuur, E. A. G. and Vogel, J. G.: Soil $CO_2$ production in upland tundra where permafrost is thawing, J. Geophys. Res. 115 G01009 DOI 1010292008JG000906 11 P, 115 [online] Available from: https://www.fs.usda.gov/treesearch/pubs/38935 (Accessed 5 July 2018), 2010.

Nowinski, N. S., Taneva, L., Trumbore, S. E. and Welker, J. M.: Decomposition of old organic matter as a result of deeper active layers in a snow depth manipulation experiment, Oecologia, 163(3), 785–792, doi:10.1007/s00442-009-1556-x, 2010.

Ping, C. L., Jastrow, J. D., Jorgenson, M. T., Michaelson, G. J. and Shur, Y. L.: Permafrost soils and carbon cycling, SOIL, 1(1), 147–171, doi:10.5194/soil-1-147-2015, 2015.

Schädel, C., Schuur, E. A. G., Bracho, R., Elberling, B., Knoblauch, C., Lee, H., Luo, Y., Shaver, G. R. and Turetsky, M. R.: Circumpolar assessment of permafrost C quality and its vulnerability over time using long-term incubation data, Glob. Change Biol., 20(2), 641–652, doi:10.1111/gcb.12417, 2014.

Schuur, E. A., Vogel, J. G., Crummer, K. G., Lee, H., Sickman, J. O. and Osterkamp, T. E.: The effect of permafrost thaw on old carbon release and net carbon exchange from tundra, Nature, 459(7246), 556–559, 2009.

Vaughn, L. J. S., Conrad, M. E., Bill, M. and Torn, M. S.: Isotopic insights into methane production, oxidation, and emissions in Arctic polygon tundra, Glob. Change Biol., 22(10), 3487–3502, doi:10.1111/gcb.13281, 2016.

Wainwright, H. M., Dafflon, B., Smith, L. J., Hahn, M. S., Curtis, J. B., Wu, Y., Ulrich, C., Peterson, J. E., Torn, M. S. and Hubbard, S. S.: Identifying multiscale zonation and assessing the relative importance of polygon geomorphology on carbon fluxes in an Arctic Tundra Ecosystem, J. Geophys. Res. Biogeosciences, 2014JG002799, doi:10.1002/2014JG002799, 2015.

---

## Author Comment (AC2) · 26 Jul 2018

**Response to anonymous Referee #2**

We welcome this reviewer's thoughtful comments and suggestions and have provided responses to individual comments below. Our responses describe changes that will be submitted in the revised manuscript.

General comments

*The authors present soil respiration quantity and 14CO2 signature as well as soil pore 14CO2 data of high-latitude soils during the thawing seasons of 2012 -2014. Acquiring this type of data in these regions is not trivial and the data provides some interesting insights into the seasonally and spatially variable carbon respiration in this region. The data is provided under the given link and the R codes used for the graphs and modeling are provided on github. Overall, the data and presented insights are valuable and merit publication.*

*Some aspects regarding the sampling documentation, blank analyses, statistics as well as data use in models could be improved upon in order for this dataset in order to reach its full potential. Specifically, as micro topographic variability was identified as a key driver for variability in 14CO2 trends, a visualization and quantification of that topography would be helpful.*

In response to this comment and a comment from Reviewer #1, we will add a schematic diagram of polygon microtopography to the revised manuscript. In the study site section, we have also added values for the percent coverage of each polygon type across the Utqiaġvik region land surface, as estimated by a classification algorithm in Lara et al. (2014). Quantitative estimates of the percent cover of each polygon type are unavailable for the specific study region, but we have provided a relative ranking of the coverage of each polygon type, based on the classification in Wainwright et al. (2015).

*Furthermore, respiration data of soil carbon is directly linked to the solid bulk soil carbon – however, that there is no data on this pool.*

The explicit focus of this analysis was the carbon contributing to respiration. Because the most fast-cycling carbon pool dominates the decomposition flux but generally makes up only a small portion of the total soil carbon pool, $\Delta^{14}C$ of ecosystem or soil respiration is not directly linked to that of bulk soil carbon (Trumbore, 2000). For this reason, given the high cost of radiocarbon analyses, we chose to focus this study's analytical effort on $CO_2$ data. We do agree, however, that bulk soil radiocarbon values provide a metric describing carbon stabilization within soil, and that a comparison between spatial patterns in the $CO_2$ flux and bulk soil values could potentially be informative. In a separate study using laboratory incubations of soil cores, we measured $\Delta^{14}C$ of shallow and deep bulk soil increments collected in September 2014 from the same locations as this study's measurements. Based on this reviewer's suggesiton, we have added an additional panel to figure 2 showing these bulk soil $\Delta^{14}C$ values from each polygon type. These bulk soil $\Delta^{14}C$ measurements will be added to the data archive as well.

*Although analytical precision data was provided, no data on procedural blanks were provided. This type of sampling and measurements take many intermediate steps, and if available it would be good to have procedural blank data in addition to just the OXAs (e.g., Hanke et al., 2017). Especially because the samples were processed not in one batch but rather over a number of years (2014-2017).*

The paper referenced by the reviewer addresses compound-specific radiocarbon analysis, in which the small-size samples are particularly prone to contamination by non-sample carbon during the various processing steps. In such cases, procedural blanks are important for quantifying contamination during the sample preparation steps. With environmental $CO_2$ samples as in this study, our understanding is that procedural blanks are not typically used. In place of blanks (which would not yield a carbon quantity large enough for the radiocarbon analysis system used), we evaluated our air sampling canisters with repeated leak testing and an ethanol-derived certified $CO_2$ reference gas standard with empirically determined $^{14}$C (Airgas, USA) to confirm that no contamination was introduced during sample storage. Additionally, the vacuum line used for $CO_2$ purification was tested with USGS coal and oxalic acid II standards for contamination or leakage. Standards used were well characterized with respect to $^{14}$C. Our understanding is that these quality control measures are standard and sufficient for this sample and analysis type. Details on the standards used have been added to the methods section of the revised manuscript.

*Regarding statistics, there are some points were statistical terms such as standard deviation is used when n=2, or box plots with quartiles are implemented when n = 3 or 4. This is could be seen as misleading and result in misinterpretation. See specific comments for suggestions how this can be improved.*

We have responded to these comments in the line-by-line responses below.

*Considering that this is a journal aimed at geo-data and use of this data in other studies, the authors could add details regarding which specific types of models could benefit from this data. This could improve how this data is used in the future stages.*

Radiocarbon measurements have potential applications for both bottom-up and top-down models of the carbon cycle. These uses include constraining the decomposition rate parameters used in bottom-up models with empirical distributions of pool-specific carbon cycling rates, evaluating the abilities of such models to produce realistic decomposition rate distributions, and determining temporal variations in $^{14}$C of ecosystem respiration, important for top-down (inverse) models that use atmospheric radiocarbon time series to evaluate net $CO_2$ sources and sinks. Such models are quite broad in scope, and could include any models with pool-specific decomposition rate parameters (which are otherwise challenging to determine in situ), models with a built-in $^{14}$C output (which can be used as a direct benchmarking method), or top-down models that use radiocarbon in atmospheric $CO_2$.

These applications were briefly mentioned in the manuscript's introduction, but we have expanded this section at the reviewer's request. In the revised manuscript, this passage now reads: "Radiocarbon measurements of soil respiration may be particularly useful for models of the carbon cycle. First, any bottom-up carbon cycle models that employ pool-specific respiration rate parameters may benefit from radiocarbon-derived soil carbon pool structures and decomposition rates. Without the radiocarbon tracer, such metrics are challenging to measure in situ, particularly where carbon cycling rates are slow. Second, by including a $^{14}$C calculation for carbon pools and fluxes, such models can use radiocarbon measurements in surface $CO_2$ emissions and soil pore gas as a benchmarking tool. Additionally, the radiocarbon signature of ecosystem respiration can be used to constrain the terrestrial signal in top-down carbon cycle analyses (He et al., 2016; Randerson et al., 2002)."

Specific Comments

*Regarding the novelty of the data and methods, the data is novel and focuses on a region which is undergoing rapid environmental change. Data is based on established sampling and analytical methods.*

*Regarding the data being used in the future: The authors refer to the data being used in 'models', but could perhaps provide more detail. At some point, they refer to a publication (He et al., 2016) that uses Earth System models, but I can imagine that there could be a larger scope of the implementation of this data in models.*

We have addressed this comment in our previous response. We would also welcome any suggestions from the community for additional model applications.

*One major piece of information which is missing - which is likely important for these types of models – is the size and 14C signature of the solid soil carbon. It is unknown how the amount of respired carbon relates to this –presumably much larger – pool.*

As discussed above, in response to this comment we have provided $\Delta^{14}$C values for bulk soil carbon collected in September 2014 from the same locations as the ecosystem respiration samples. We stress, however, that the radiocarbon values of bulk soil carbon are not necessarily well correlated with those of the respired fraction.

*One additional aspect which can be improved upon in order to increase the value of this data is an improved description of the morphology of the micro topographic features that were sampled. The authors underline that micro topography has a strong impact on 14C signature of respired carbon, but there are no pictures or sketches. I would recommend adding photographs or a sketch showing the features. Furthermore, in order to be able to extrapolate this data, it would be helpful to know how often which type of polygon types occurs. That way, the signal could be averaged for this region.*

As discussed in our response above, we have provided additional information on polygon microtopography in the manuscript's introduction. This includes background information

on how controls on carbon cycling relate to polygon morphology, as well as a distribution of polygon types across the land surface at our site.

*The materials and methods are described in sufficient detail and the R code and data themselves are provided. The article itself is appropriate for the publication of this dataset because it clearly describes how the samples was acquired and data was measured.*

*As mentioned previously, if there is data on the bulk carbon quantity and signature from other papers, it could be valuable to cite and integrate that. Presently it is not present in the paper.*

As discussed above, we will add bulk soil carbon data to the manuscript and data archive.

Data quality and accessibility

*The data is accessible & codes are available on git. There are even some 2-pool model codes on git which I believe are not used in the paper. Error estimates could be improved upon. Mostly only the analytical error is provided. I am missing the procedural blank, which is especially important as the samples were measured over three years. The sample sizes for 14C measurements are also not provided. Smaller samples are more susceptible to contamination (Hanke et al., 2017). Ideally, there would be more sample replicates. However, considering the difficulty of the sampling and the significant cost of 14C measurements it makes sense that no extensive replication done in this case.*

We agree with the reviewer that susceptibility to contamination increases with smaller sample sizes.  To address this concern, we will add sample sizes to the data archive.  These sample sizes are all greater than 0.1 mg C, with the exception of one small sample (0.044 mg C; soil pore-space $CO_2$ collected from HC1-center in August 2012).  As discussed at length in our response above, our understanding is that procedural blanks are not typically used for this type of $^{14}C$ sample.  Instead, standard analyses and leak testing were used as alternative quality control measures.

*There is also a potential bias in the soil respiration data that has not yet been addressed. The authors state that when the soil was water saturated, it was not possible to measure pore CO2. It is very likely that there is a difference in carbon decomposition speed between water saturated and non-water saturated soils. Therefore, if only the non-saturated soils soil pore CO2 is measured, there could be a bias in the interpretation. Authors should address this in the discussion.*

We agree with this astute point raised by the reviewer.  We were cautious in our interpretation not to make statistical inferences from our data because of sampling issues such as this.   This issue was briefly addressed in the final paragraph of section 3.2, but we agree that this consideration deserves greater attention.  Accordingly, we have added several sentences to this paragraph, which now reads: "Due to sampling limitations, $^{14}C$ data from soil pore-space $CO_2$ are available from only a subset of polygon types, positions, sapling dates, and depths.  No samples were collected from polygon rims, and

only few samples were obtained where soils were saturated (Table 2).  For this reason, our dataset does not capture the full range of microtopographic variations in deep soil decomposition rates and controls. Soil temperature profiles, soil pore-space oxygen availability, soil organic matter concentration and chemistry, and vegetation composition, rooting distribution, and productivity all influence microbial activity and vary among profiles and with time in the thawed season (Lipson et al., 2012; Olivas et al., 2011).  In particular, these controls on decomposition dynamics likely differ between saturated and unsaturated soils, due both to the direct effects of saturation and to covarying effects of microtopography.  For this reason, radiocarbon values presented in this manuscript cannot be scaled across the full range of environmental variation present at the site.  To better characterize the relationships between these variables and soil carbon decomposition rates, further measurements are needed of $^{14}$C in soil pore-space DIC and $CO_2$, across spatial, seasonal, and hydrological gradients.”

*Dataset quality. Data seems in the range of what can be expected from these types of cold soils.*

Dataset presentation

*The graphs can be improved upon on a number of points, especially concerning the usage of some statistical terms, as detailed in the section technical comments. Note that box plots are not designed for data where n = 3.*

*Nor is standard deviation applicable if n = 2. Standard deviation is a measure to see what the spread of data is, but with n= 2 you don’t really have a spread.*

See below for responses to each of these points.

Publication length

*The length of the paper is good, the figures and table are appropriate. If the authors provide more information of the sampling and micro topography, the dataset could be used in the future after reading the paper & downloading the data*

Technical comments

*Page 1 line 11: Specify model*

We have changed this to read, “models of the carbon cycle.”  We chose not to go into greater detail in the abstract, but have added further discussion of model applications in the conclusion section.

*Line 12: Specify use of D14C*

We have added the units $\Delta^{14}C_{Reco}$ and $\Delta^{14}C_{CO2p}$.

*Line 25: You state that the flux is critical – is there any quantification?*

This introductory sentence is meant to emphasize that the variability in ecosystem-atmosphere flux over space and time makes it challenging to quantify, both empirically and through models. Accordingly, we feel that in this context, it is more appropriate to leave this as a qualitative statement rather than provide quantification either at the global scale or for a specific location.

*Page 2 lines 12-15: Sentence is very long and not clear*

In response to the reviewer's comment, we have broken this sentence into two shorter sentences and have changed the wording. This passage now reads, "There, an estimated 1300 Pg of soil carbon has been protected from decomposition by cold temperatures and often frozen or anoxic conditions. With climate change, these controls on decomposition rates are expected to change as soils warm, hydrological changes occur, and permafrost degradation intensifies."

*Study site section: If possible, add a sketch of polygon and how dominant each polygon type is.*

This comment has been addressed at two points above.

*Page 5 lines 2-3: 'measurements lacking a linear range were not included in the dataset' Do you mean samples that fall outside of the linear range, or complete measurement series?*

For clarity, this has been changed to "curves lacking a clear linear range."

*Line 14: Low-concentration $CO_2$ measurements were omitted except when value was clearly not-atmospheric. Does that mean that the excluded samples all had atmospheric signature?*

Yes, the $\Delta^{14}$C values of the excluded samples ranged from +13.6 ‰ to +59.4 ‰, all indicating a large percentage of carbon with a near-atmospheric radiocarbon signature. Because the $CO_2$ yields for these samples were at or below the expected yields at atmospheric concentration and pressure, we believe there was a high likelihood of leakage around the soil probe during sampling. As discussed in the paper, subsurface $\delta^{13}$C of $CO_2$ can vary considerably at this site due to methane production and oxidation, so we were unable to calculate the degree of atmospheric carbon in soil pore-space samples. For this reason, we believed the conservative approach of excluding low-concentration samples was appropriate. While we believe this choice was based on sound reasoning, the reviewer's comment has led us to use more precise language in this passage. In the revised manuscript, this now reads: "With soil pore-space radiocarbon data, we omitted measurements with $CO_2$ yields below the expected yield for atmospheric measurements due to possible leakage and atmospheric contamination during sampling"

*Line 16: Pore CO₂ was not measurement for water saturated soils.  This may bias final interpretations.  Please address*

As discussed above, we took care in our interpretation not to draw statistical inferences because of this and other potential sampling biases.  In response to the reviewer's comment and to emphasize this consideration, we have changed this sentence to read, "For this reason, the final sample set represents only a subset of depths and sampling locations and is not a random sample of the entire landscape."  Additionally, we have added a passage discussing this point to the discussion.

*Page 6 line 11: Tr or mean the of carbon in plants is either assumed to be 0 or 5 year.  Whilst cited sources give a which wider range.  Even decomposing pine needles in a temperate zone are generally 8 years old.  In such cold environments with slow growth rates one could expect even older ages of plant-derived carbon into the soil.  Is this assumption realistic?  Please add some further justification.*

We agree with the reviewer that a portion of plant-derived carbon resides for considerably longer than 5 years within plant tissues before being incorporated into soil organic matter.  As the reviewer points out, the sources cited in our manuscript indicate a range of as long as 15 years for certain plant species and organs.  We recognize as well that in these cold environments, adaptations of select species may lead to even longer carbon residence times within certain living plant tissues.  However, we emphasize that the transit time described by the 0-5 year $T_R$ range approximates the flux-weighted mean of *all* carbon passing through living plant tissues.  In Barrow and other Arctic sites, the majority of aboveground vegetation tissues turn over annually.  Annual leaf growth, for example, accounts for the majority of total annual production in Arctic tundra sites, even though it amounts to a much smaller fraction of the standing biomass (Shaver and Kummerow, 1991).  Turnover of this aboveground leaf litter (with a < 1-year transit time) thus represents a large proportion of the plant-derived carbon flux. In addition, some non-structural photosynthates enter the soil within one or several days of fixation, contributing an additional source of carbon with very low $T_R$ values.  Given the importance to total annual carbon assimilation of leaf production and rapid photosynthate transport to the soil, we believe the range of 0-5 years is an appropriate—and conservative—$T_R$ estimate.

We recognize that more justification for this range could have been provided in the manuscript, so we have included the following explanation in the revised manuscript: "The mean residence time of carbon in plants reflects a mixture of materials with varying transfer rates.  In Arctic vegetation dominant at our site, some photosynthates enter the soil within 1 day of fixation (Loya et al., 2002); leaf tissues with transit times of < 1 year represent the majority of annual production (Shaver and Kummerow, 1991); roots and shoots are estimated to live for ranges of 3-7 and 1-8 years respectively (Chapin III et al., 1980; Shaver and Billings, 1975); and rhizomes and stem bases, the longest-lived of belowground tissues, have estimated turnover times of 2.7-15.6 years (Dennis, 1977).  Overall, the mean residence time of carbon in plants ($T_R$) represents the flux-weighted average of various pools.  Given the large percentage of carbon flux dedicated to annual

leaf production and rapid transport to soils, we estimate that across plant organs, plant species, and seasons, the mean value of $T_R$ lies between 0 and 5 years."

*Lines 21-25: When two turnover times are equally likely (Graven, 2015), it could also be helpful to have measured the bulk soil signature. If this is indeed not available, this is the best fix possible, but it is not ideal.*

We respectfully disagree with the reviewer's comment. Even when bulk soil radiocarbon values are available, the present-day bulk soil turnover time offers little to no information about the turnover time of the rapidly decomposing fraction. Please see Trumbore *et al.*, (2000) and Torn *et al.*, (2009) for further background and discussion of this issue.

*Page 7 line 14: When describing the effects of the polygons and microtopography, a sketch or image would be helpful for the understanding of the reader.*

Based on the reviewer's suggestion, we will include a sketch of polygon microtopography to the revised manuscript's introduction.

*Page 8 line 10: Authors state that pre $^{14}CO_2$ becomes increasingly negative with depth, but this is not the case for three profiles (as mentioned later in the text on line 18)*

This section has been restructured in a way that clarifies the fact that three profiles do not display the depth trend seen in the other profiles. This paragraph now begins, "In most soil profiles, $\Delta^{14}C_{CO2p}$ became increasingly negative with depth in the soil. This depth trend is similar to that seen in the bulk soil carbon profiles, indicating that the cycling rates of both bulk soil organic matter and the decomposing carbon fraction tend decrease with depth in the soil. Because frozen or near-freezing temperatures slow decomposition from the deep active layer throughout the majority of summer, this pattern is what would be expected in the absence of vertical mixing or rapid contributions of fast-cycling carbon at depth. In contrast with this general trend, soil pore-space $CO_2$ from three soil profiles (HC3-trough, HC3-center, and FC2-center became enriched in radiocarbon near the permafrost table (Fig. 4)…"

*Page 8 line 13: Author state that there are minimal contributions from plant-respired carbon. Did they do a mass-balance? Or is this a qualitative statement? Please specify.*

This is a qualitative statement based on the fact that $\Delta^{14}C_{CO2p}$ values were negative—in most cases highly negative—with the modeled mean age of respired carbon ranging from 410 to 3350 years. Without known isotopic end-members for the plant-respired and soil-respired fractions, it is not possible to do a mixing model (mass-balance) to quantify these contributions. In theory, it is possible to perform incubations with vegetation and soils from a range of depths and use the $^{14}C$ and $^{13}C$ abundances of the evolved $CO_2$ to constrain this type of mixing model. In practice, however, this approach cannot be used reliably at our site where spatially- and temporally-variable methane processes influence $^{13}C$ abundance in soil pore-space $CO_2$ (Vaughn et al., 2016). This issue is discussed in the methods section (page 5, lines 16-18 of the original manuscript).

At the reviewer's request, we added a sentence to the end of this paragraph that clarifies this point. This paragraph now ends, "Detecting and characterizing the decomposition of older, deeper soil organic carbon requires direct measurements of soil pore-space $CO_2$. Such measurements provide a qualitative indicator of old carbon decomposition; as with surface $CO_2$ effluxes, proportional or absolute contributions from distinct carbon pools cannot be calculated without well-resolved vertical distributions of $^{14}C$ and $^{13}C$ source pools, as well as gas transport within the profile."

*Lines 13-17: These statements support Figure 4. For better flow, maybe move towards end of section*

We agree with the reviewer that these statements support Figure 4, and have added a reference to this figure in the text. We have chosen not to move this statement to the end of the section, but instead have changed the wording to improve flow: "These consistently negative values indicate that pore-space $CO_2$ was derived primarily from older soil organic matter, with minimal contributions from plant-respired carbon or fast-cycling soil organic carbon. In contrast, ecosystem respiration was generally enriched in radiocarbon relative to soil pore-space $CO_2$, even at only 10 cm depth (Fig. 4)."

*Line 19: Figure 2 should be 4*

This typo has been corrected.

*Lines 25-26: When trying to understand soil carbon stability and decomposability, it would be good to not only determine the age of respired $CO_2$ but also the age of the bulk soil carbon. If this is not available for these profiles, but similar ones, it would be good to benchmark the data.*

As discussed above, we have included bulk soil radiocarbon profiles for a subset of the locations in this dataset.

*Page 9 line 9: Authors state: "due to sampling limitations, soil profile 14C data are available from only a subset of polygon types[...] In tables 1 & 2 I see only soil CO2 measurements, no bulk. Did they mean to refer to pore CO2?*

Yes, this statement refers to soil pore-space $CO_2$. To clarify, we have changed this to read, "$^{14}C$ data from soil pore-space $CO_2$"

*Page 9 line 35: 'later in season' please be more specific*

This line has been changed to "In September and October, over half the $\Delta^{14}CO_2$ measurements differed greatly from the local atmosphere…"

*Table 1: Standard deviation with n = 2 does is not appropriate and can be seen as misleading (go back to calculation). It is supposed to give the range of sample variability, but for n = 2 that is not applicable. Consider an alternative, e.g., giving the average*

Based on the comments from Reviewer #1, we have chosen no longer to include Table 1 in the main manuscript. Instead, we have added two figure panels that summarize the soil temperature and thaw depth data. All data originally listed in Table 1 are included in the data archive, where both the mean and standard deviation of each set of measurements are listed. For consistency, this includes values for which n = 2. While the standard deviation when n = 2 is not particularly informative, we do not believe that it is statistically misleading, as it is not being used for statistical inference.

*Table 2: HC1-center: Measurements were done at different depths during different years. Or is it a typo? If not, considering the inter annual variability, is it reasonable to compare the data? Please clarify.*

This is not a typo. As stated in the text, we were unable to obtain measurements from only a subset of polygon types and positions. For clarity, we have expanded this explanation to state that samples were unavailable not only from all polygon types and positions but also from all sampling dates and depths. When presenting these data in Figure 4, we grouped measurements by soil profile *and* sampling date, in order to emphasize that measurements from different depths and years should be considered separately. In Table 2, we believe it is appropriate to include the full dataset. For clarity, we have added a line break between samples collected on different dates.

*Table 2: For HC3 center for the same depth there are large differences in 14C*

This is true. As stated above, we have added line breaks between sampling dates to clarify that these measurements were made on different dates. The observed differences in 14C are likely due not only to inter-annual variability but also to seasonal differences in soil temperature profiles, root respiration, and soil carbon decomposition rates.

*Figure 1: It appears that boxplot R are used for n = 2-5. It may not be the most appropriate way of presenting the range of data. Also add in the explanation that when R boxplots are visualized, data that falls outside the range is statistically speaking an outlier. Also indicate what the solid line in the box means. Also considering adding a striped line for the mean.*

Following the reviewer's suggestion, we have modified this figure so that it is no longer a box plot. The revised figure is a dot plot, with day of year on the x-axis instead of month. We agree that this formatting more accurately reflects the distribution of the data and the exact sampling times.

*Figure 1: Considering using this plot to highlight the important finding that during high-summer, all respiration is from year to decadal old C, whilst later in the season, whilst the deeper soil continues to warm, older (stabilized) C is lost.*

We agree with the reviewer's interpretation of this figure. We believe this finding is thoroughly addressed in the original manuscript in the second paragraph of section 3.2: "As shallow soils warm and plant activity increases in this early summer period, ecosystem respiration includes high proportions of $^{14}$C-enriched $CO_2$ from autotrophic respiration and heterotrophic decomposition of shallow, rapid-cycling soil carbon. Later

in summer and into the autumn, the balance of respiration shifts toward increased importance of deeper soil decomposition. Autotrophic respiration peaks in July or August, and decreases substantially into the fall after plants senesce (Hicks Pries et al., 2013). During the autumn season, surface soils refreeze while deep soils continue to warm (Zona et al., 2016), limiting heterotrophic respiration from shallow soils while enhancing decomposition from deeper, more $^{14}$C-depleted soil carbon pools. The effect of these changes is a seasonal shift in respiration from primarily shallow, fast-cycling source carbon pools to more deep, $^{14}$C-depleted soil organic matter." This passage follows a reference to Figure 1, but we have added an additional reference to Figure 1 to highlight the connection between the figure and this interpretation.

*Figure 2: Consider changing axis (starting at 0.3) in a to highlight differences in respiration*

We thank the reviewer for the suggestion, but prefer not to change this figure's axis. Given that the standard error of each measurement is far greater than any differences in mean respiration rates, we do not believe it is appropriate to highlight these differences.

*Figure 3: Legend should be month-year.*

This has been changed to month-year in the revised manuscript.

*Figure 3: This graph shows an interesting trend which has not been explicitly discussed in the text, but which could be valuable for the data interpretation. There could be two end member type of behaviours: High summer (2013-07), high respiration of topsoil C releases bomb-enriched carbon. Late summer (2014-09) releases a low amount of old, stabilized carbon. 2013-09 could be an intermediate type where there is mixing from both pools, providing a spread of ages and concentrations.*

To an extent, we agree with the reviewer's interpretation. This figure suggests that during high summer (July 2013), all respiration has a bomb-enriched radiocarbon signature. In September of 2014, in contrast, we observed a wide range in radiocarbon values, with the samples divided between bomb-enriched and $^{14}$C-depleted signatures. However, because respiration rates were highly variable in July 2013 and $\Delta^{14}$C values were highly variable in September 2014, we hesitate to classify these sampling periods into distinct "high respiration of bomb-enriched C" vs. "low respiration of old, stabilized C" regimes. Additionally, because one of three dates (September 2013) does not fit into either regime, we do not think the data are sufficient to support this interpretation. Instead, as discussed in the final paragraph of section 3.1, we feel that they key takeaway from this figure is that old, stabilized carbon constitutes a large percentage of the total $CO_2$ flux *only* when respiration rates are low.

*Figure 4: Colour spread is not optimal, some colors are near-identical. Considering changing. Also consider highlighting the 3 soils which some younger respired C at depth, or grouping samples by polygon type.*

Based on these comments and those from Reviewer 1, we have changed this figure substantially in the revised manuscript. Figure 4 now groups $\Delta^{14}C_p$ profiles by position within polygon and polygon type and no longer lists the specific profile ID and measurement month/year. We have also added a horizontal line at 0 representing the soil surface and moved the chamber flux data to +2 cm depth. As a consequence of these changes, the color scheme is simplified and improved.